# A vagal reflex evoked by airway closure

Michael S. Schappe[1], Philip A. Brinn[1], Narendra R. Joshi[1], Rachel S. Greenberg[1], Soohong Min[1], AbdulRasheed A. Alabi[1], Chuchu Zhang[1] & Stephen D. Liberles[1✉]

Airway integrity must be continuously maintained throughout life. Sensory neurons guard against airway obstruction and, on a moment-by-moment basis, enact vital reflexes to maintain respiratory function[1,2]. Decreased lung capacity is common and life-threatening across many respiratory diseases, and lung collapse can be acutely evoked by chest wall trauma, pneumothorax or airway compression. Here we characterize a neuronal reflex of the vagus nerve evoked by airway closure that leads to gasping. In vivo vagal ganglion imaging revealed dedicated sensory neurons that detect airway compression but not airway stretch. Vagal neurons expressing PVALB mediate airway closure responses and innervate clusters of lung epithelial cells called neuroepithelial bodies (NEBs). Stimulating NEBs or vagal PVALB neurons evoked gasping in the absence of airway threats, whereas ablating NEBs or vagal PVALB neurons eliminated gasping in response to airway closure. Single-cell RNA sequencing revealed that NEBs uniformly express the mechanoreceptor PIEZO2, and targeted knockout of *Piezo2* in NEBs eliminated responses to airway closure. NEBs were dispensable for the Hering–Breuer inspiratory reflex, which indicated that discrete terminal structures detect airway closure and inflation. Similar to the involvement of Merkel cells in touch sensation[3,4], NEBs are PIEZO2-expressing epithelial cells and, moreover, are crucial for an aspect of lung mechanosensation. These findings expand our understanding of neuronal diversity in the airways and reveal a dedicated vagal pathway that detects airway closure to help preserve respiratory function.

Sensory neurons that constitute the interoceptive nervous system relay information to the brain from vital organs in the body[5]. Within the airways, sensory neurons provide essential feedback to control breathing, promote gas exchange, protect the airways through cough and laryngeal guarding reflexes, detect pathogens to induce sickness, and elicit perceptions of breathlessness, also known as dyspnoea or air hunger[1,2,6–8]. Neuronal surveillance of the airways enables the detection of life-threatening inefficiencies in gas exchange, which can be a characteristic of cardiopulmonary disease. In response, neural circuits trigger compensatory reflexes such as gasps (quick and deep inspirations also called augmented breaths or sighs) to reopen closed airways, bring air into the lungs and relieve respiratory distress[9–13]. Gasps are triggered by many airway threats, including hypoxia, bronchospasm, pulmonary congestion and thoracic compression, precede voluntary behaviours such as speaking and singing, and are often the first and last breaths of life. Neural circuits in the brain have been identified that coordinate the gasping motor response[13], but sensory pathways that lead to gasping, and their mechanisms of action, require additional exploration.

The vagus nerve provides the major sensory innervation of the airways, and classical studies have described airway neurons as rapidly adapting mechanoreceptors, slowly adapting mechanoreceptors or chemosensitive C fibres[1,2,5]. Recent single-cell expression profiling and genetic approaches have revealed a richer diversity of vagal neurons in the larynx, trachea and lungs[6,14–17]. Notably, the sensory properties and functions of many vagal neurons with particular transcriptional signatures and/or airway terminal morphologies are undefined[2,6,14,16], which suggests that additional airway-to-brain reflexes remain uncharted.

Mechanoreceptors that detect lung stretch are perhaps the best-studied airway sensory neurons. In 1868, Hering and Breuer reported that mechanical inflation of the airways causes a reflexive inhibition of breathing or apnoea[18], now termed the Hering–Breuer inspiratory reflex. Airway stretch receptors are slowly adapting mechanoreceptors activated with each inspiration during tidal breathing and directly sense lung distension through the mechanosensory ion channel PIEZO2 (refs. 6,19–21). Hering and Breuer also reported physiological responses to decreases in airway volume[18], and Adrian later observed lung deflation responses in single unit recordings of vagal afferents[19]. Deflation responses were not observed during normal expiration and were subsequently ascribed to polymodal nociceptors, including trachea-enriched C fibres, which also detect cough-evoking irritants, juxtacapillary fibres in the pulmonary vasculature (so-called J fibres) and/or rapidly adapting mechanoreceptors also activated by high threshold stretch[1,22–24]. It has remained unclear whether dedicated receptors for airway closure exist in the conducting airways of the lung. If so, their functions and mechanisms of action have remained unknown.

## Responses to airway closure in the mouse

We first sought to characterize physiological responses to airway closure in the mouse, a tractable model system that enables genetic experiments for mechanistic study. In other animals, airway closure triggers gasping and reflexive increases in inspiration, with some variability across species and paradigm reported[25–28]. We used several

[1]Department of Cell Biology, Howard Hughes Medical Institute, Harvard Medical School, Boston, MA, USA. ✉e-mail: Stephen_Liberles@hms.harvard.edu

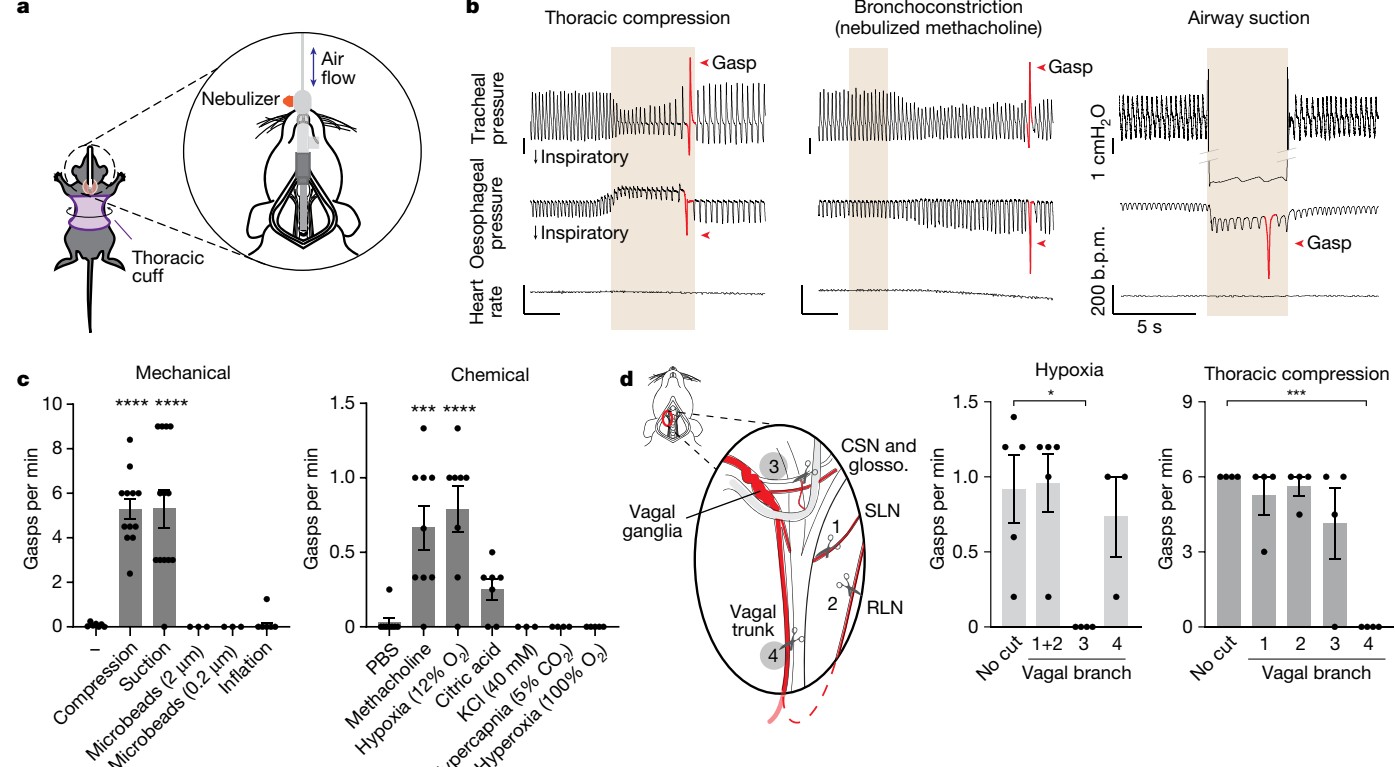

**Fig. 1 | A vagal gasping reflex to airway closure. a**, Cartoon depicting the application of airway challenges. **b**, Representative physiological measurements before, during and after airway closure (tan shading). b.p.m., beats per minute. **c**, Quantification of gasp frequency in response to the indicated stimuli. Data are mean ± s.e.m., with dots indicating the average per animal across three trials. $n$ (left to right) = 7, 12, 13, 3, 3 and 6 (mechanical); 8, 9, 8, 7, 3, 4 and 5 (chemical). Significance determined by one-way analysis of variance (ANOVA) with Bonferroni post hoc test: mechanical, $F_{5,38} = 15.61$, $P < 0.0001$; – versus compression, $P < 0.0001$; – versus suction, $P < 0.0001$; chemical, $F_{6,37} = 9.399$, $P < 0.0001$, PBS versus methacholine, $P = 0.0004$; PBS versus hypoxia,

$P < 0.0001$. **d**, Quantification of gasp frequency (right) to hypoxia (10% oxygen) or thoracic compression following transection of nerve branches as numbered in the cartoon (left). Data are mean ± s.e.m., with dots indicating the average per animal across three trials. $n$ (left to right) = 5, 5, 4 and 3 (hypoxia); 4, 4, 4, 4 and 4 (thoracic compression). One-way ANOVA with Bonferroni post hoc test: hypoxia, $F_{3,13} = 5.030$, $P = 0.0157$; no cut versus nerve branch 3, $P = 0.0157$; thoracic compression, $F_{4,15} = 11.04$, $P = 0.0002$, no cut versus nerve branch 4, $P = 0.0002$. CSN and glosso., glossopharyngeal nerve, including the carotid sinus nerve (CSN).

methods to decrease functional airway volume in the mouse: (1) airway compression by inflating a cuff surrounding the thorax; (2) airway suction by applying negative pressure within the trachea; and (3) broncho-constriction by tracheal delivery of nebulized methacholine (Fig. 1a). Physiological changes were recorded by measuring tracheal pressure (which reports the frequency and magnitude of each inspiration and exhalation), oesophageal pressure (as a proxy for intrathoracic pressure) and heart rate. Each method of airway closure triggered reflexive gasps (Fig. 1b), with a stereotypical pattern characterized by a power-ful, deep inspiration and subsequent rapid exhalation (defined by a >50% increase in expiration compared with the previous and subse-quent breath). Gasps were also associated with increased activity of intercostal muscles and the diaphragm (Extended Data Fig. 1a,b), as measured by electromyography. The frequency of gasping depended on the depth of anaesthesia, and under urethane anaesthesia, gasping was rarely observed at baseline but was routinely observed following airway compression (Extended Data Fig. 1c). Gasping was also trig-gered by hypoxia (10–12% O₂), to a lesser extent by nebulized citric acid and not by other stimuli, including inhaled particulates (microbeads), hypercapnia or hyperoxia (Fig. 1c). Mechanical stimuli evoked a higher gasp frequency than chemical stimuli tested, and this is probably due, at least partially, to a shorter time delay to first gasp after introduction of the stimulus (Extended Data Fig. 1d). We observed that thoracic compression, under these conditions, promoted additional changes in lung mechanics, including decreased lung compliance through chest

wall restriction (Extended Data Fig. 1e). Onset of airway compression reduced airway pressure and respiratory rate (Extended Data Fig. 1f), presumably because increased intrathoracic pressure provides domi-nant suppression over any compensatory inspiratory drive, but had no effect on blood oxygen saturation (Extended Data Fig. 1f–h).

Gasping responses to thoracic compression persisted following transection of several vagus nerve branches, including the superior laryngeal nerve (SLN), the recurrent laryngeal nerve (RLN) and the glossopharyngeal nerve. Responses were instead abolished after transection of the vagus nerve trunk below the SLN departure point (Fig. 1d), a procedure that eliminates vagal fibres below the larynx, including those to the lung. By comparison, gasps and increases in tidal volume induced by hypoxia persisted after transection of the vagus nerve trunk below the SLN, but instead were lost after transec-tion of the glossopharyngeal nerve, consistent with a role for carotid body chemosensation[29]. Together, these findings indicate that airway closure, induced by airway compression, suction or bronchoconstric-tion, induces a gasping reflex in the mouse through the vagus nerve.

## Vagal activity during airway closure

Vagal fibre recording studies have produced conflicting data about whether airway deflation decreases activity of airway stretch neurons, activates dedicated neurons and/or activates polymodal neurons that also detect other stimuli, such as chemical threats or high-threshold

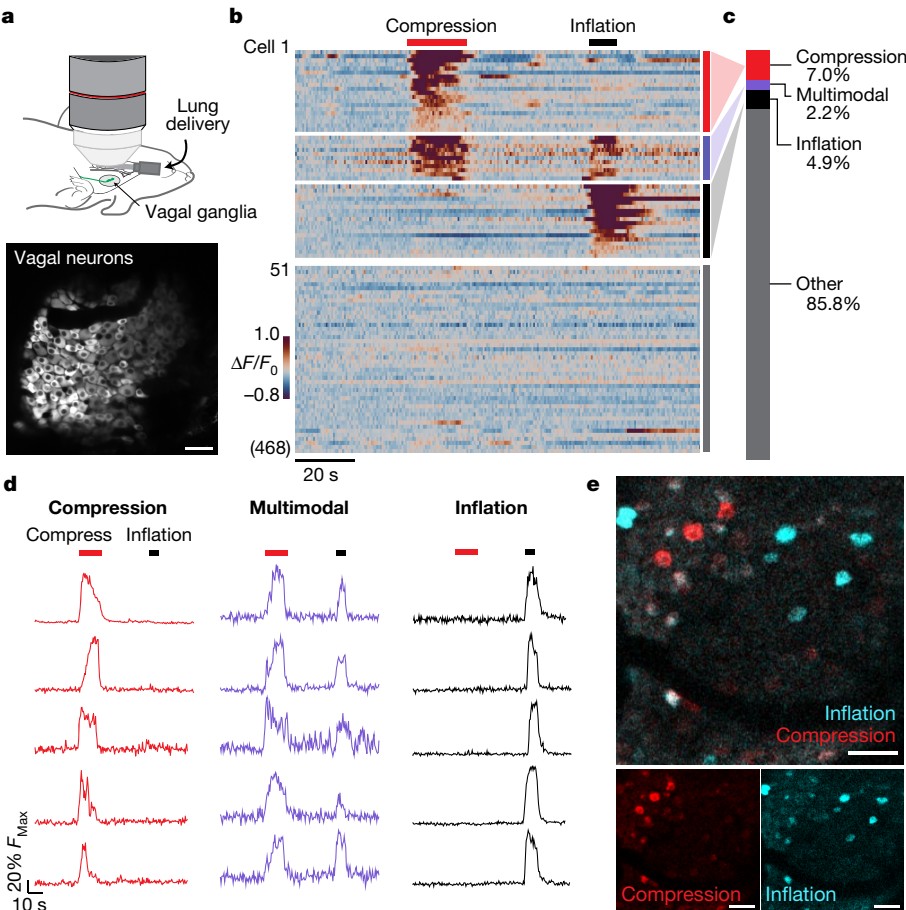

**Fig. 2 | Imaging neuronal responses to airway closure. a**, Cartoon depicting vagal ganglion imaging (top) and a two-photon image (bottom) of SALSA fluorescence in vagal ganglia of *Vglut2-ires-cre;lsl-SALSA* mice. **b**, Heatmap depicting vagal sensory neuron calcium responses ($\Delta F/F$ colour coded, 468 imaged neurons, 2 representative mice) to airway compression (red bar) and airway inflation (black bar). All 55 responsive and some randomly selected non-responsive neurons are shown. **c**, Quantification of neurons responsive to only compression (red, 91 out of 1,303 neurons), only inflation (black, 64 out of 1,303), both (purple, 29 out of 1,303) or neither (grey, 1,119 out of 1,303) across 1,303 neurons, 7 mice. **d**, Representative traces of ratiometric GCaMP6f fluorescence from individual vagal neurons responding to the indicated stimuli. **e**, Representative images of maximal GCaMP6f fluorescence in vagal ganglion neurons during airway compression (blue) and inflation (red). Scale bars, 50 μm (**a**,**e**).

airway stretch[19,22,23,30]. Here we used in vivo calcium imaging within vagal ganglia[6,20] to investigate vagal responses to airway closure. Vagal ganglion imaging enables a parallel analysis of real-time responses in >100 individual neurons per experiment. We used the progeny of *lsl-SALSA* mice mice, which express the calcium reporter GCaMP6f-tdTomato (also called SALSA[31]) from a Cre-dependent allele, crossed with *Vglut2-ires-cre* mice, in which Cre recombinase is expressed in all vagal sensory neurons. We imaged vagal ganglia while connections to the lungs were preserved (Fig. 2). As we previously observed[20], airway stretch evoked calcium transients in a small group of vagal sensory neurons (7.1%, 93 out of 1,303 neurons, 7 mice) that express PIEZO2 (refs. 6,20,21). Compressing the airways by inflating a thoracic cuff triggered acute calcium responses in 9.2% of vagal sensory neurons (120 out of 1,303 neurons, 7 mice). Most airway closure receptors (75.8%, 91 out of 120) did not respond to airway stretch, but some (24.2%, 29 out of 120) responded weakly. Airway suction activated a smaller group of vagal sensory neurons (5.1%, 25 out of 495), most of which also responded to airway compression. Nebulized methacholine also stimulated some vagal neurons (Extended Data Fig. 2c,d), which largely overlapped with compression-sensing neurons (60.7%, 54 out of 89) but not neurons selective for lung inflation (33 out of 89 responded to compression but not inflation, 21 out of 89 responded to both compression and inflation and 3 out of 89 responded to inflation but not compression) (Extended Data Fig. 2d and Supplementary Video 1). The percentages

of neurons that responded to different stimuli across trials were generally conserved across mice (Extended Data Fig. 2a,b). Together, these findings indicate that airway compression acutely activates a subset of vagal sensory neurons, with the major cohort unresponsive to airway stretch, which suggests that there is a dedicated vagal pathway for detecting airway closure.

## Vagal PVALB neurons mediate gasping

Vagal ganglia contain dozens of molecularly distinct sensory neuron types[6,15,17,32,33], so we asked which sensory neurons mediate airway-closure-induced gasping. First, we used optogenetics to activate various vagal sensory neurons and measured reflexive gasping behaviour. We expressed the light-activated ion channel channelrhodopsin-2 from a Cre-dependent allele (*lsl-ChR2*) in different vagal sensory neuron types using Cre driver mice, including *P2ry1-ires-cre*, *Pvalb-t2a-cre*, *Crhr2-ires-cre*, *Npy2r-ires-cre*, *Gpr65-ires-cre* and *Vglut2-ires-cre* mice. As previously reported[6,14,20,34], we then stimulated vagal sensory neurons by shining light on the vagus nerve trunk, ganglion or particular nerve branches (Fig. 3a). Activating all vagal sensory neurons in *Vglut2-ires-cre;lsl-ChR2* mice evoked reflexive gasps, with a frequency of 6.0 gasps per min of illumination, whereas activating CRHR2, NPY2R, GPR65 or other neuron types did not (Fig. 3b). Optogenetic stimulation of PVALB or P2RY1 neurons similarly caused fictive gasping, with PVALB

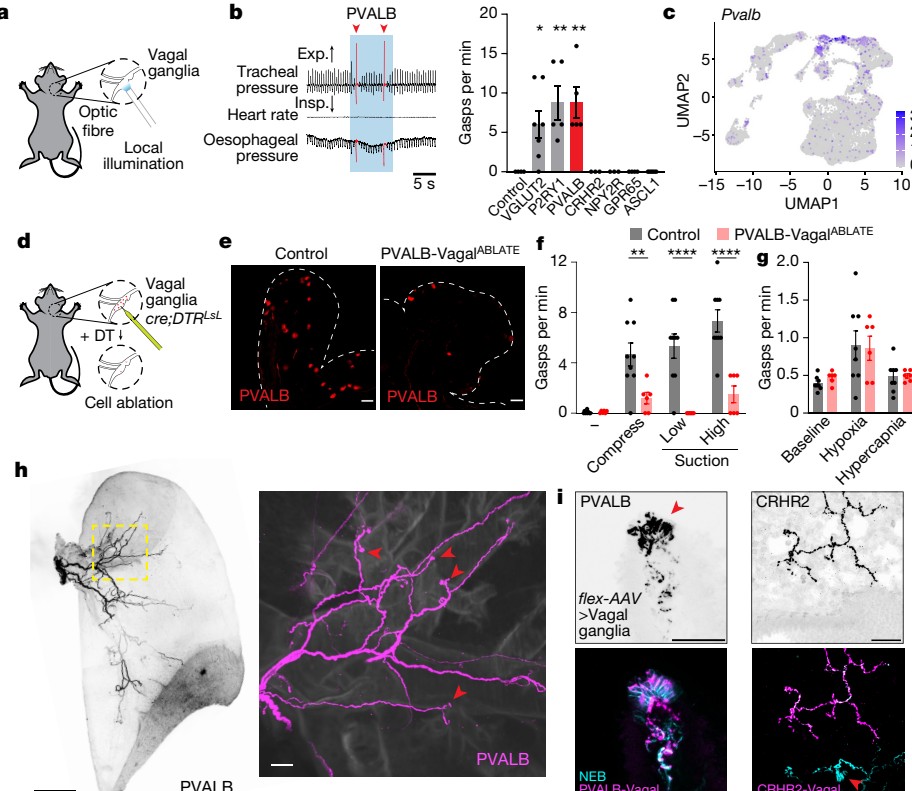

**Fig. 3 | Vagal gasping neurons innervate NEBs. a**, Cartoon depicting vagal optogenetics. **b**, Representative physiological changes (left) and gasp quantification (right) from optogenetic stimulation (blue shading) of indicated vagal neurons. Exp., expiration; Insp., inspiration; red triangle, gasp. Data are mean ± s.e.m., with dots indicating the average per animal across three trials. $n$ (left to right) = 4, 7, 5, 5, 3, 3, 4 and 6. One-way ANOVA (Bonferroni post hoc): $F_{7,29}$ = 7.912, $P$ < 0.0001; pairwise comparisons to control: VGLUT2, $P$ = 0.0399; P2RY1, $P$ = 0.0022; PVALB, $P$ = 0.0022, others are not significant (NS). **c**, Uniform manifold approximation and projection (UMAP) plot from published single-cell transcriptome data of vagal and glossopharyngeal sensory ganglia[6] indicating *Pvalb* expression (purple shading, natural log scale). **d**, Cartoon depicting targeted vagal neuron ablation. **e**, Representative image of native tdTomato fluorescence in vagal ganglia of *Pvalb-t2a-cre;lsl-DTR;lsl-tdTomato* mice with or without DT injection. **f**, Gasp frequency in PVALB-Vagal[ABLATE] mice or control Cre-negative DT-injected littermates. Dashed lines indicate vagal ganglia. Data are the mean ± s.e.m., with dots indicating the average responses of individual animals across 4 trials per mouse. $n$ = 9 (control) and 6

(PVALB-Vagal[ABLATE]) mice. Two-way ANOVA (Bonferroni post hoc): condition × genotype $F_{3,39}$ = 7.066, $P$ = 0.0007; pairwise comparisons of control versus PVALB-Vagal[ABLATE]: –, NS; compress, $P$ = 0.0066; low suction, $P$ < 0.0001; high suction, $P$ < 0.0001. **g**, Gasp frequency by whole-body plethysmography in PVALB-Vagal[ABLATE] mice or control Cre-negative DT-injected littermates. Data are the mean ± s.e.m., with dots indicating individual animals. $n$ = 8 (control) and 6 (PVALB-Vagal[ABLATE]) mice. Two-way ANOVA (Bonferroni post hoc): condition × genotype $F_{2,24}$ = 0.1435, $P$ = 0.8671; pairwise comparisons: control versus PVALB-Vagal[ABLATE]: all NS. **h**, Representative low-magnification (left) and zoomed-in (right, yellow box) images of native tdTomato fluorescence in a iDisco-cleared lung from a *Pvalb-t2a-cre;Vglut2-Flpo* mouse containing a Cre-dependent and Flp-dependent tdTomato reporter (*inter-Ai65*). Red arrowheads indicate nerve terminals. **i**, Representative images of tdTomato (magenta) and NCAM1 (cyan) immunohistochemistry in lung from *Pvalb-t2a-cre* (left) and *Crhr2-ires-cre* (right) mice injected in vagal ganglia with *AAV-flex-tdTomato*. Red arrowheads indicate nerve terminals. Scale bars, 50 μm (**i**), 100 μm (**e**,**h** (right)) or 1 mm (**h**, left).

predominantly expressed in subsets of vagal P2RY1 neurons. Activation of P2RY1 but not PVALB neurons also caused swallowing, as previously reported[6] (Extended Data Fig. 3a,b,c). In *Pvalb-t2a-cre;lsl-ChR2* mice, gasping was evoked by illumination of vagal ganglia or the vagal trunk distal to the SLN departure point, consistent with a role for lung afferents (Extended Data Fig. 3d). Gasps were evoked by ganglion illumination after transection of the trunk, which indicated a role for sensory neurons that transmit the information to the brain (Extended Data Fig. 3d). *Pvalb* is enriched in three clusters of vagal sensory neurons, only one of which also expresses *Olfr78*. We generated *Olfr78-p2a-cre* mice, and optogenetic stimulation of vagal OLFR78 sensory neurons also evoked gasping behaviour (Extended Data Fig. 3e). These experiments pinpoint transcriptome-defined vagal neuron subtypes that mediate gasping (Fig. 3c), and these neurons are distinct from P2RY1 neurons that mediate swallowing[6].

We then asked whether vagal PVALB neurons would respond to airway closure. We crossed mice to contain a constitutive neuronal GCaMP reporter (*Snap25-GCaMP6s*), *Pvalb-t2a-cre* and a Cre-dependent

fluorescent reporter (*lsl-tdTomato*) to distinguish Cre-positive and Cre-negative neurons. We then performed calcium imaging in vagal ganglia as described above. We observed that PVALB neurons accounted for most neurons that responded to both airway suction and compression (13 out of 14, 92.9%; Extended Data Fig. 3f). Moreover, airway closure activated 24.8% of vagal PVALB neurons, a result consistent with transcriptomics analysis indicating that gasp-promoting vagal sensory neurons represent 23.1% (161 out of 698) of all PVALB neurons (Extended Data Fig. 3e).

*Pvalb-t2a-cre* mice are an effective tool to mark some neurons that evoke gasps and respond to airway closure, so we asked whether ablating vagal PVALB neurons eliminated gasping to airway closure. Vagal PVALB neurons were engineered to express the human receptor (DTR) for diphtheria toxin (DT) using a Cre-dependent allele[35], and DT was injected directly into vagal ganglia (resulting in PVALB-Vagal[ABLATE] mice; Fig. 3d). This DT-based approach has been effectively used for targeted cell ablation in the vagus nerve, and results in efficient ablation of Cre-positive cells in vagal ganglia but not Cre-negative cells

or Cre-positive cells in other locations[6,34,36]. We verified similarly efficient removal of vagal PVALB neurons using this approach compared with control mice (Fig. 3e and Extended Data Fig. 4a), which were age-matched Cre-negative littermates injected with DT. Control mice displayed normal gasping responses to airway compression, but airway compression-evoked gasps were lost in PVALB-Vagal[ABLATE] mice (Fig. 3f) as well as in P2RY1-Vagal[ABLATE] mice (Extended Data Fig. 4b,c). PVALB-Vagal[ABLATE] mice also exhibited decreased gasping to airway suction and methacholine, but normal rates of hypoxia-evoked gasping compared with littermates (Fig. 3f,g and Extended Data Fig. 4e). No significant changes in tidal volume or breathing rate during relaxed breathing were observed (Extended Data Fig. 4d), but suction-induced increases in inferred tidal volume were reduced in PVALB-Vagal[ABLATE] mice (Extended Data Fig. 4f). Finally, the Hering–Breuer inspiratory reflex was normal in PVALB-Vagal[ABLATE] mice (Extended Data Fig. 4g), which indicated that reflexes evoked by airway stretch and airway closure are mediated by distinct vagal neuron types. Taken together, activating vagal PVALB neurons evokes gasps, whereas ablating them eliminates gasping to airway closure.

## Airway closure neurons appose NEBs

Various terminal types have been proposed in the airways[1,2,6,14,17], but the location and structure of neuronal terminals sensitive to airway closure are unknown. We previously developed a genetic approach to trace the terminals of Cre-defined vagal sensory neurons[14], and here performed similar experiments in *Pvalb-t2a-cre* mice to genetically mark airway-closure-sensing terminals in the lungs. We injected the vagal ganglia of *Pvalb-t2a-cre* mice with an adeno-associated virus (AAV) harbouring a Cre-dependent reporter allele encoding tdTomato (*AAV-flex-tdTomato*). Alternatively, we visualized labelled neurons directly in *Pvalb-t2a-cre;Vglut2-Flp* mice containing a Cre-dependent and Flp-dependent tdTomato reporter gene. We observed dense innervation of the lungs in *Pvalb-t2a-cre* mice (Fig. 3h), with all visualized lung axons displaying characteristic candelabra terminals in airway epithelium. PVALB candelabra terminals uniformly apposed clusters of specialized lung secretory cells termed NEBs, which were visualized by immunostaining for neural cell adhesion molecule 1 (NCAM1) (Fig. 3i). We observed innervation of 62% of lung NEBs by vagal sensory neurons in general using a Cre-independent AAV-based tracer (130 out of 209 NEBs, 18 mice), and innervation of a similar percentage of NEBs (60%) by vagal PVALB neurons (Extended Data Fig. 4h,i). PVALB neurons innervated NEBs as densely as vagal P2RY1 neurons[14]. Moreover, NEBs were not innervated by other airway vagal sensory neurons marked in many other Cre lines, including *Crhr2-ires-cre*, *Gabra1-ires-cre*, *Npy2r-ires-cre*, *Mc4r-ires-cre* and *Gpr65-ires-cre*. By comparison, vagal CRHR2 neurons promoted apnoea and instead innervated near terminal bronchioles and alveoli, but did not innervate NEBs. Thus, vagal PVALB neurons mediate airway closure responses and densely innervate NEBs in the lung.

## NEBs mediate gasping to airway closure

NEBs are clusters of rare epithelial cells (<1% of all epithelial cells) located predominantly near branch points of the major conducting airways in the lung[37]. The dense innervation of NEBs suggests that NEBs are primary sensory cells analogous to taste cells or Merkel cells, other excitable epithelial cells that communicate with neurons. It has been proposed that NEBs detect a variety of stimuli, including hypoxia, acid and mechanical stretch[38–42], and they have been linked to neuroimmune signalling and airway pathophysiology[43–46]. Our observations that NEBs are densely and specifically innervated by vagal PVALB neurons raised the possibility that they may be involved in detecting airway closure.

We developed intersectional genetic approaches to gain selective access to NEBs in vivo. We combined a Flp allele (*Nkx2.1-Flpo*) that broadly marks pulmonary epithelial cells[47] (Fig. 4a) and a Cre allele that marks NEBs but not other pulmonary epithelial cells (*Piezo2-ires-cre*; see single-cell analysis of NEBs below). We also generated *Ascl1-creER;Nkx2.1-Flpo* mice (followed by tamoxifen administration in adults), as *Ascl1* is a classical marker for NEBs[48]. In both *Piezo2-ires-cre;Nkx2.1-Flpo* mice and in *Ascl1-creER;Nkx2.1-Flpo* mice, expression of Cre-dependent and Flp-dependent reporter genes were driven with high selectivity and efficiency in NEBs (Fig. 4b). Selective labelling of NEBs within the lung was verified by co-staining for NCAM1, with NCAM1 also labelling nearby neuronal fibres that were not labelled by intersectional genetic tools. In both allele combinations, reporter genes were not observed in any other cell types of the lung, neurons of vagal ganglia, spinal cord or dorsal root ganglia, or in cells of the carotid body or aortic arch (Extended Data Fig. 5). Solitary neuroendocrine cells in the larynx and trachea were effectively labelled in *Ascl1-creER;Nkx2.1-Flpo* mice but generally not labelled (<1%) in *Piezo2-ires-cre;Nkx2.1-Flpo* mice (Extended Data Fig. 5b). Additionally, we noted reporter expression in the thyroid and the hippocampus of *Ascl1-creER;Nkx2.1-Flpo* mice but not in the thyroid or brain of *Piezo2-ires-cre;Nkx2.1-Flpo* mice. Based on these data, we conclude that *Piezo2-ires-Cre;Nkx2.1-Flpo* mice (hereafter termed NEB[INTER] mice) are an effective tool for NEB-selective genetic manipulations.

Next, we used chemogenetic approaches to activate NEBs in freely behaving mice, and measured the effects on respiratory physiology and behaviour by whole-body plethysmography (Fig. 4c). We used an intersectional reporter (*inter-Gα_q-DREADD*) that drives expression of a designer $G\alpha_q$-coupled receptor for the synthetic agonist clozapine-*N*-oxide (CNO)[49]. CNO elicited calcium transients in NEBs in live lung slice preparations from $G\alpha_q$-DREADD-expressing mice (tamoxifen-treated *Ascl1-creER;lsl-SALSA;lsl-Gα_q-DREADD* mice; Extended Data Fig. 6a), which confirmed that chemogenetic tools can directly stimulate NEBs. In freely behaving mice, activating NEBs with CNO caused reflexive gasping in *NEB[INTER];inter-Gα_q-DREADD* (termed NEB[Gαq-DREADD]) mice (Fig. 4d). Breathing rate and tidal volume were not significantly altered (Extended Data Fig. 6b,c), but NEB activation changed the respiratory pattern, with accelerated inspiration and increased peak inspiratory flow (Fig. 4e). Similar results were observed whether NEBs were intersectionally targeted with *Piezo2-ires-cre* or *Ascl1-creER* alleles (Extended Data Fig. 6d,e). Gasps induced by NEB activation displayed a similar characteristic pattern to gasps induced by airway closure or to spontaneous sighs (Extended Data Fig. 6f,g), which suggests a common motor command circuit. CNO-induced gasps in NEB[Gαq-DREADD] mice persisted under urethane anaesthesia and after transecting the SLN, RLN or glossopharyngeal nerves, but were abolished by transection of the vagus nerve trunk after the SLN departure point (Extended Data Fig. 7a), a result consistent with a role for lung afferents. In addition to respiratory changes, chemogenetic activation of NEBs in freely behaving mice caused apparent respiratory distress, with a characteristic hunching posture and immobility (Supplementary Video 2 and Extended Data Fig. 7b,c), as might be expected for a mouse experiencing air hunger.

We then used intersectional genetic approaches to ablate NEBs. We crossed NEB[INTER] mice to contain an intersectional allele driving expression of DTR (*inter-DTR*)[50]. DT administration in *NEB[INTER];inter-DTR* mice effectively ablated 87% of NEBs (termed NEB[ABLATE] mice) compared with DT-injected control littermates lacking Cre recombinase (Fig. 4f). We first asked how NEB ablation affected vagal responses to airway closure, as measured by whole nerve electrophysiology. Vagal responses to airway suction were reproducibly observed across multiple trials in control mice, but were absent in NEB[ABLATE] mice. By comparison, vagal responses to airway inflation were observed in both control and NEB[ABLATE] mice, a result consistent with a model in which distinct sensory pathways mediate airway closure and inflation (Fig. 5g). Furthermore, NEB[ABLATE] mice failed to gasp in response to airway closure caused by airway compression, suction or methacholine, whereas normal gasping

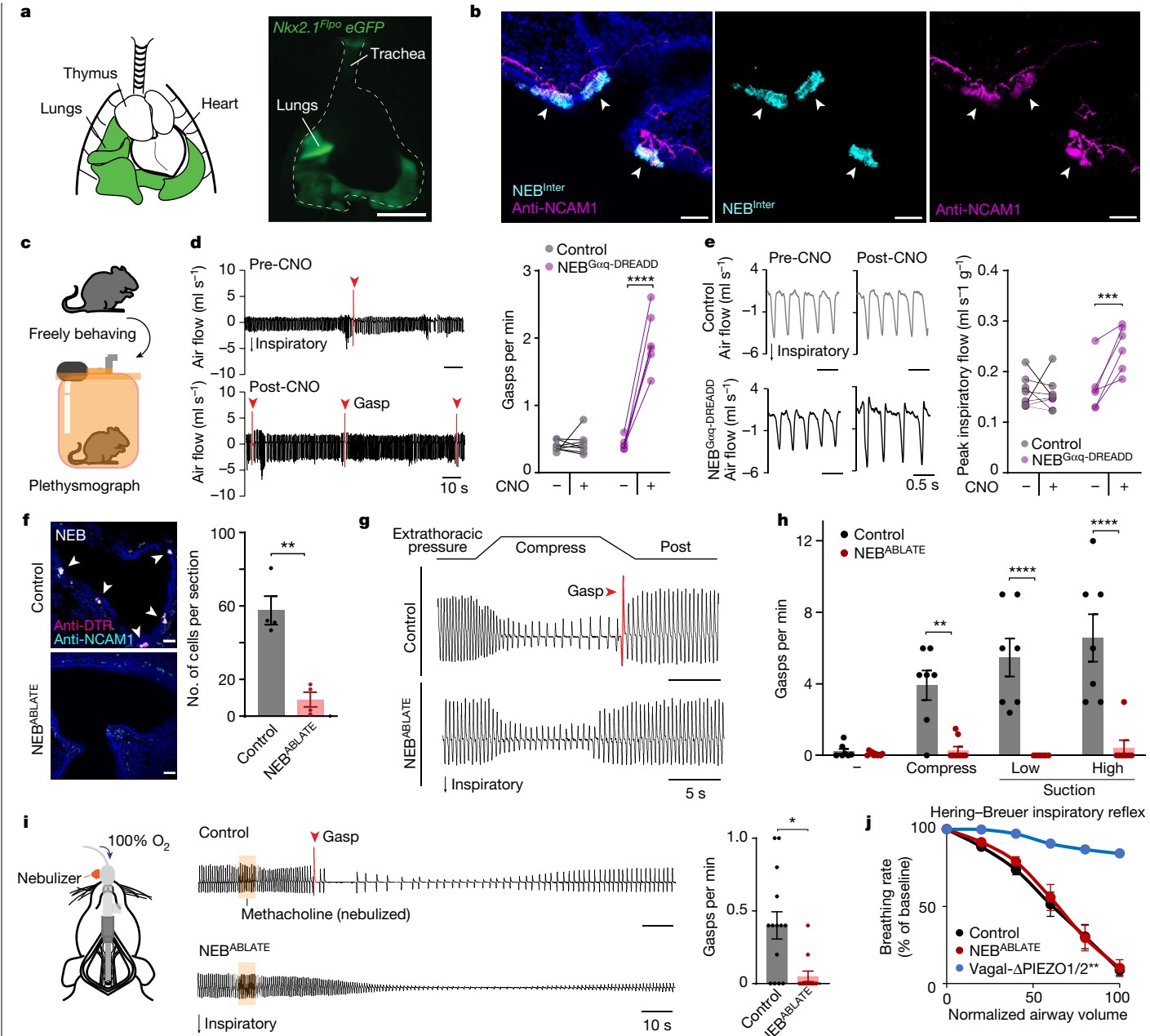

**Fig. 4 | NEBs mediate gasps. a**, Cartoon (left) and image of native thoracic reporter fluorescence (right). **b**, mCherry (cyan) and NCAM1 (magenta) immunochemistry in lung cryosections of NEB^Gαq-DREADD-mCherry mice. **c**, Cartoon of whole-body plethysmography. **d**, Pneumotachographs (left) and gasp quantification (right) in freely behaving NEB^Gαq-DREADD mice and Cre-negative control mice before and 10 min after CNO. Dots and lines indicate individual animals $n$ = 8 (control) and 6 (NEB^Gαq-DREADD) mice. Two-way ANOVA (Bonferroni post hoc): condition (CNO) × genotype $F_{1,12}$ = 88.38, $P$ < 0.0001; adjusted $P$ value (pre-CNO versus post-CNO): control, NS; NEB^Gαq-DREADD mice, $P$ = 0.0001. **e**, Zoomed-in pneumotachographs (left) and peak inspiratory flow quantification (right) as in **d**. Same mice as **d**; two-way ANOVA (Bonferroni post hoc): condition (CNO) × genotype $F_{1,12}$ = 11.43, $P$ = 0.0055; adjusted $P$ value (pre-CNO versus post-CNO): control, NS; NEB^Gαq-DREADD mice, $P$ = 0.0022. **f**, NEBs visualized (left) and counted (right) in lung cryosections from NEB^ABLATE mice and control Cre-negative DT-injected littermates. Data are the mean ± s.e.m., with dots indicating the averages per animal across four sections. $n$ = 4 mice per group. Unpaired $t$-test, $P$ = 0.0014. **g**, Representative tracheal pressure traces. **h**, Gasp frequency. Data are the mean ± s.e.m., with dots indicating the average of 3–4 trials per animal. $n$ = 7 mice per group. Two-way ANOVA (Bonferroni post hoc): condition × genotype $F_{3,36}$ = 11.75, $P$ < 0.0001; pairwise comparisons (control versus NEB^ABLATE, left to right): NS, $P$ = 0.0028, $P$ < 0.0001, $P$ < 0.0001. **i**, Cartoon (left), representative tracheal pressure traces (middle) and gasp quantification (right) 5 min after methacholine exposure (orange bar). Data are the mean ± s.e.m., with dots indicating individual animals. $n$ = 14 (control) and 12 (NEB^ABLATE) mice. Unpaired $t$-test, $P$ = 0.003. **j**, Breathing rates following lung inflation to assess the Hering–Breuer inspiratory reflex. Data are the mean ± s.e.m. $n$ = 9 (control), 7 (NEB^ABLATE) and 3 (Vagal-ΔPIEZO1/2) animals. Two-way ANOVA (Bonferroni post hoc): condition (normalized airway volume) × genotype $F_{10,80}$ = 7.464, $P$ < 0.0001; all pairwise comparisons after inflation (control versus NEB^ABLATE), NS; all pairwise comparisons (control versus Vagal-ΔPIEZO2), **$P$ < 0.0031. Scale bars, 25 μm (**f**), 50 μm (**b**) or 5 mm (**a**).

responses were preserved in control littermates (Fig. 4g–i). Breathing rate and tidal volume were unchanged in NEB^ABLATE mice during relaxed breathing, but we noted that NEB^ABLATE mice had decreased

compensatory changes in tidal volume during and after airway compression (Extended Data Fig. 8). NEB^ABLATE mice also had decreased lung compliance and inspiratory capacity, which indicated that NEBs

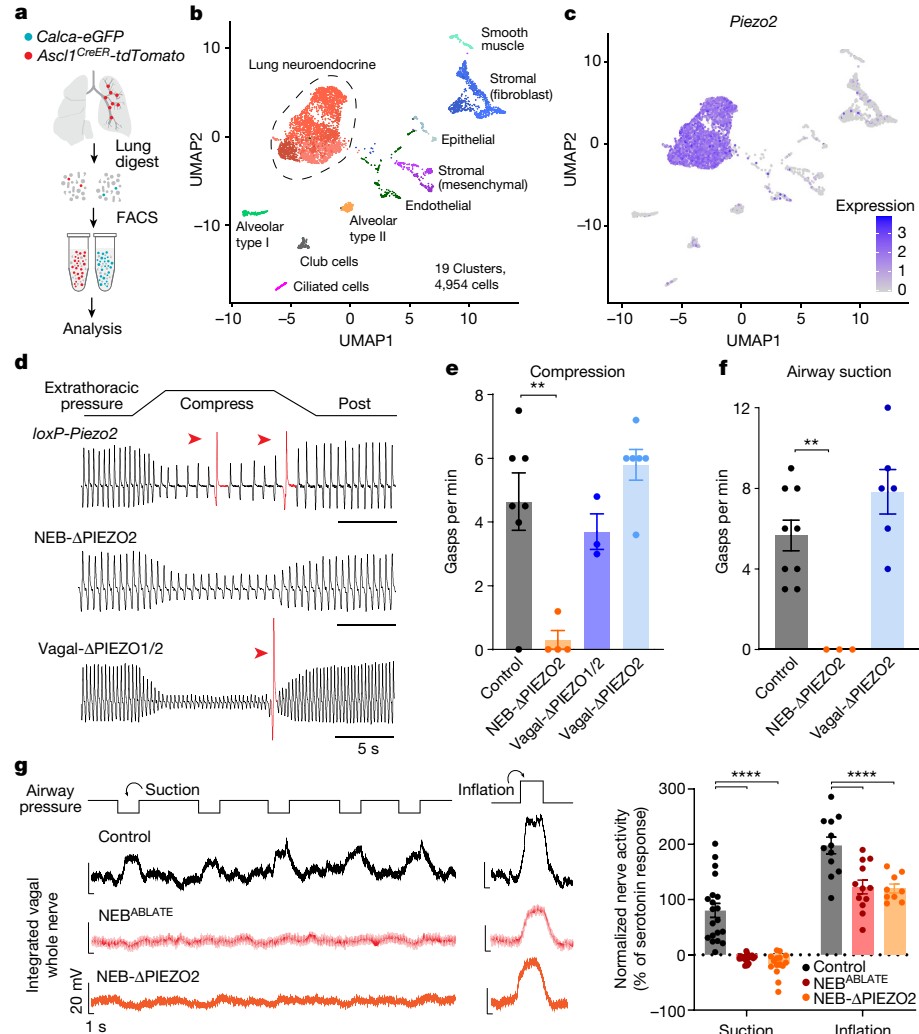

**Fig. 5 | Airway closure responses require PIEZO2 in NEBs. a**, Cartoon depicting NEB isolation strategy. **b**, UMAP plot indicating cell types from single-cell RNA sequencing data. **c**, UMAP plot of *Piezo2* expression (purple shading, natural log scale) in single-cell transcriptome data from **b**. **d**,**e**, Representative traces of tracheal pressure (**d**) and quantification (**e**) of compression-evoked gasps in the indicated mice. Data are the mean ± s.e.m., with dots indicating individual animals. $n = 7$ (control), 4 (NEB-ΔPIEZO2), 3 (Vagal-ΔPIEZO1/2) and 6 (Vagal-ΔPIEZO2) mice. One-way ANOVA with Bonferroni post hoc test: condition (compress) × genotype $F_{3,16} = 9.424$, $P = 0.0008$; control versus NEB-ΔPIEZO2, $P = 0.0021$; all other pairwise comparisons to control, NS. **f**, Quantification of suction-evoked gasps in the indicated mice. Data are the mean ± s.e.m., with dots indicating individual animals. $n = 9$ (control), 3 (NEB-ΔPIEZO2) and 6 (Vagal-ΔPIEZO2) mice. One-way ANOVA with Bonferroni post hoc test: condition (compress) × genotype $F_{2,15} = 11.76$, $P = 0.0008$; control versus NEB-ΔPIEZO2, $P = 0.0063$; all other pairwise comparisons to control, NS. **g**, Representative recordings (left) and quantification (right) of vagal nerve activity during airway suction (low) or inflation. Data are the mean ± s.e.m., with dots indicating individual trials. $n = 5$ suction trials, 3 inflation trials per mouse from 4 control, 4 NEB^ABLATE, and 3 NEB-ΔPIEZO2 mice. Two-way ANOVA with Bonferroni post hoc test (genotype): $F_{2,82} = 44.42$, $P < 0.0001$; control versus NEB^ABLATE or NEB-ΔPIEZO2, $P < 0.0001$; all other pairwise comparisons, NS.

have an active role in maintaining normal mechanical properties of the lung. Spontaneous sighs were still displayed, but with a 36% reduction in frequency, a result consistent with the presence of NEB-independent pathways for sigh generation[13,29]. The Hering–Breuer inspiratory reflex was normal in NEB^ABLATE mice across a range of airway distension volumes (Fig. 4j), so NEBs are dispensable for PIEZO2-mediated detection of airway inflation[21]. Taken together, NEBs are not required for normal breathing but are essential to maintain normal lung mechanics and to induce gasps when the airways are challenged by closure.

## NEB-localized PIEZO2 and airway closure

Physiological responses to airway closure require NEBs, which suggests that at least some NEB cells express an airway-closure-activated mechanosensor. NEBs have been proposed to sense a diversity of airway cues[38,45,51], but the underlying sensory receptors that mediate airway

closure are unclear. Here we analysed the transcriptomes of individual NEB cells to search for mechanosensory proteins that detect airway closure in the lungs.

NEBs are sparse cells (constituting around 0.4% of lung epithelial cells), so we used genetic tools to collect them for single-cell RNA sequencing (Fig. 5a). Immune-cell-depleted suspensions of lung cells were made from mice in which NEBs were fluorescently labelled (*Calca-eGFP* or *Ascl1-creER;lsl-tdTomato*), and NEBs were enriched by fluorescence-activated cell sorting (FACS), with only a minority of non-NEB cells also captured and included for comparative analysis. Individual cells were then encapsulated in nanolitre-sized droplets using the 10x Genomics platform, and cell-barcoded cDNA was synthesized and sequenced. Transcriptomes were obtained from 4,954 cells, including 2,975 NEB cells (defined based on known markers), and an atlas of cell types was created using unsupervised bioinformatics analysis (Fig. 5b and Extended Data Fig. 9a–c). Pathway analysis revealed

that NEBs had enriched expression of neurotransmitters and genes associated with synapse formation (Extended Data Fig. 9d). Depending on the stringency of cell-type assignment criteria, a few NEB cell types could be subclustered[44,52]. Immunochemistry previously revealed that the mechanosensor PIEZO2 is expressed in at least some NEB cells[21]. Notably, single-cell RNA sequencing data here revealed uniform *Piezo2* expression by NEB cells (Fig. 5c), a result similarly observed by *Piezo2* RNA in situ hybridization (Extended Data Fig. 10a) and by genetic marking in *Piezo2-ires-cre* mice (Fig. 4b).

PIEZO2 acts directly in other vagal sensory neurons to detect airway stretch and mediate the Hering–Breuer inspiratory reflex[21] and, together with PIEZO1 in other vagal sensory neurons, to detect changes in blood pressure underlying the baroreceptor reflex[34,53]. Notably, PIEZO2 is expressed in NEBs and some PVALB neurons, which raised the questions of whether PIEZO2 acts in the sensation of airway closure and, if so, where it acts. Global *Piezo2* knockout is lethal soon after birth owing to respiratory distress[21], so we used approaches for cell-selective elimination of *Piezo2*. We previously described *Phox2b-cre;flox-Piezo2* mice and *Phox2b-cre;flox-Piezo1;flox-Piezo2* mice[21,53], in which PIEZO2 or both PIEZO1 and PIEZO2 are deleted from nodose/petrosal ganglia but not NEBs (herein termed Vagal-ΔPIEZO2 mice and Vagal-ΔPIEZO1/2 mice, respectively). We additionally generated *Ascl1-creER;flox-Piezo2* mice, in which PIEZO2 expression is removed from NEBs but not vagal or glossopharyngeal sensory neurons after tamoxifen administration (NEB-ΔPIEZO2 mice). Selective and effective deletion was verified by *Piezo2* RNA in situ hybridization (Extended Data Fig. 10a,b).

Next, we measured responses to airway closure following *Piezo2* deletion in either NEBs or vagal afferents (Fig. 5d–f and Extended Data Fig. 10c). Knockout of *Piezo2* in NEBs eliminated gasping responses to airway closure evoked by airway compression, airway suction or nebulized methacholine. Gasping evoked by hypoxia persisted in NEB-ΔPIEZO2 mice (Extended Data Fig. 10d), which suggested that there are distinct pathways for detecting airway closure and hypoxia, with hypoxic gasps presumably due to carotid body chemosensation[29]. Furthermore, whole nerve electrophysiology revealed that vagal responses to airway closure were absent in NEB-ΔPIEZO2 mice (Fig. 5g). By contrast, gasping to airway compression and suction persisted after knockout of PIEZO channels in vagal sensory neurons (in both Vagal-ΔPIEZO2 and Vagal-ΔPIEZO1/2 mice), even though these mice had a dysfunctional Hering–Breuer inspiratory reflex[21] (Figs. 4j and 5g and Extended Data Fig. 4g). Similar to NEB[ABLATE] animals, NEB-ΔPIEZO2 mice displayed normal eupneic breathing, spontaneous sighs with a slightly decreased frequency, decreased inspiratory capacity and decreased lung compliance (Extended Data Fig. 10e,f). Together, these findings indicate that NEBs first sense airway closure through PIEZO2 and then communicate to vagal PVALB neurons, which relay the information to the brain to ultimately evoke reflexive gasping behaviour.

## Discussion

Sensory neurons densely innervate the airways to ensure the constancy of breathing throughout life. Recent studies have revealed a diversity of transcriptome-defined vagal sensory neurons in the airways[6,15–17], many of which are 'orphan neurons' without known functions or sensory properties. Here we ascribe a function to an orphan vagal neuron type, finding that it is required for protective gasping responses to airway closure.

Airway closure threatens pulmonary function, alters breathing patterns, evokes sensations of dyspnoea or breathlessness and promotes gasping. The forceful exchange of air that occurs during a gasp helps open the airways and restore lung capacity[9]. Here we used intersectional genetics, live cell imaging, anatomical mapping, targeted neuronal manipulations and physiology to reveal the following findings: (1) a vagal pathway that senses airway closure and promotes gasping in response; (2) the localization of gasp-promoting vagal terminals at

NEBs; (3) a required role for NEBs in airway-closure-induced gasping; (4) a cell atlas of NEB expression; and (5) a required role for NEB-localized PIEZO2 in airway closure responses. Together, these results define a vagal reflex and underlying sensory mechanism that involves PIEZO2 and epithelial cell-to-neuron communication for sensing and responding to airway closure.

Gasping is also evoked by hypoxia through the carotid body[29,54], which suggests the convergence of different sensory stimuli onto gasping control circuits in the brainstem, which involve around 200 neurons in the preBötzinger complex that express bombesin-like neuropeptide receptors[13]. The preBötzinger complex receives input from key respiratory control nuclei, including the nucleus of the solitary tract (NTS)[55], which is the predominant target of vagal and glossopharyngeal afferents in the brain. The NTS displays spatial organization[56], with pulmonary P2RY1 neurons terminating in lateral NTS regions associated with breathing control[14]. In future studies, it will be interesting to understand how NTS neurons responsive to airway stretch and airway closure differentially communicate with downstream breathing control circuits[55,57–59].

NEBs were previously proposed to communicate with neurons[37], but their sensory functions remained unclear. Here we used intersectional genetics to selectively eliminate NEBs from the mouse and observed that NEB loss does not change normal breathing but instead eliminates gasping evoked by airway closure. Single-cell RNA sequencing revealed uniform expression of PIEZO2 in NEBs, and selective knockout of *Piezo2* in NEBs eliminated airway-closure-induced gasps. Although some heterogeneity was observed across NEB single-cell transcriptomes, the uniform expression of PIEZO2 raises the possibility that they serve a homogeneous mechanosensory function in detecting airway closure. We propose that NEBs are similar to Merkel cells in the skin involved in touch sensation in that they are PIEZO2-utilizing epithelial cells that communicate with peripheral sensory neurons[3,4]. However, there are residual touch responses following *Piezo2* knockout in Merkel cells, whereas airway closure responses reported here are absent following *Piezo2* knockout in NEBs. In addition to the key mechanosensory role defined here, it is possible that NEBs are polymodal sensory cells that detect other gasp-inducing stimuli; transcriptome data should provide a framework for understanding the comprehensive functions of this poorly understood cell type. NEB frequency is dynamic[51], with the number of NEBs reported to increase in various pulmonary diseases, including neuroendocrine cell hyperplasia of infancy, chronic obstructive pulmonary disorder and asthma, each of which present with symptoms of dyspnoea[8,60]. It will be interesting to investigate whether NEBs underlie physiological responses to airway closure in humans and induce the sensation of air hunger.

Over 150 years ago, classical studies revealed the first mechanosensory reflex of the vagus nerve in which increases in airway volume cause apnoea or a reduction in breathing. Here we described the workings of another mechanosensory reflex of the vagus nerve within the lungs and showed that this reflex is triggered by airway closure and leads to gasping behaviour. One model to explain these findings is that airway closure prevents dissipation of inspiratory pressure across the entire pulmonary tree, which leads to local pressure increases in conducting airways where NEBs are enriched. A model involving mechanosensation at airway bottlenecks would also explain why some classically defined deflation receptors are also activated by large lung inflations, with transient activity (apparent rapid adaptation) potentially explained by a short-lived pressure increase in large conducting airways that soon disperses throughout the lungs. PIEZO2 acts in NEBs to detect airway closure in the lung, whereas PIEZO2 instead acts directly in vagal sensory neurons to detect lung inflation without utilizing NEBs, indicating a heterogeneity of mechanosensory pathways within the lungs. This dedicated vagal pathway for detecting airway closure is therefore distinct from classical lung stretch receptors in that airway closure responses involve different vagal neurons, a specialized

mechanosensory structure involving terminals at NEBs and a distinct physiological response. The existence of distinct classes of airway mechanoreceptors presumably enables precise, bidirectional control of breathing.

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

# Methods

## Animals

All animal procedures followed ethical guidelines outlined in the NIH Guide for the Care and Use of Laboratory Animals, and all procedures were approved by the Institutional Animal Care and Use Committee at Harvard Medical School. Animals were maintained under constant environmental conditions ($23 \pm 1\,°C$, $46 \pm 5\%$ relative humidity) with food and water provided ad libitum in a 12-h light–dark cycle. All studies used adult male and female mice (6–24 weeks) in comparable numbers from mixed genetic backgrounds. All CreER mice and control littermates received tamoxifen (Sigma, T5648, $100\,\mathrm{mg\,kg^{-1}}$, intraperitoneally, sunflower oil, twice 48 h apart) at least 10 days before further experiments. Mice containing Cre-dependent and Flp-dependent DTR alleles were a gift from M. Goulding, and *Calca-eGFP* mice were purchased (GENSAT, RRID: MMRRC_011187-UCD). For *Pvalb-t2a-cre* mice, only female Cre mice were used for husbandry owing to reported germline recombination in male breeders (Jackson Laboratory, 012358); male and female offspring were used for experiments. *Olfr78-p2a-cre* mice were generated by pronuclear injection of Cas9 protein, CRISPR sgRNAs targeting the *Olfr78* locus 3′ UTR and a single-strand DNA template containing a *p2a-cre* gene cassette with 150 bp homology arms into C57BL/6 embryos. Knock-in pups were screened by PCR analysis, and correct expression of the transgene was verified by RNA in situ hybridization. All Cre driver lines used were viable and fertile, and abnormal phenotypes were not detected. Genotyping primers for *Olfr78-p2a-cre* mice were GGATGGTAAGGGTCACGTGTT (wild-type allele primer), CCGTTTTGGAAACAGCCTGG (*p2a-cre* allele 5′ primer) and TGCGAACCTCATCACTCGTT (*p2a-cre* allele 3′ primer), with differentially sized PCR products for the wild-type allele (192 bp) and knock-in allele (562 bp) in two separate reactions. All other mice were purchased from the Jackson Laboratory or made in-house and then deposited into the Jackson Laboratory: *Ascl1-creERT2* (012882), *Nkx2.1-ires-Flp* (028577), *Piezo2-eGFP-ires-cre* (027719), *inter-Gα$_q$-DREADD* (26942), *lsl-SALSA* (31968), *lsl-TdTomato* (007914), *snap25-Gcamp6s* (25111), *lsl-ChR2* (012569), C57BL/6J (000664), *lsl-Gα$_q$-DREADD* (026220), *loxP-Piezo2* (027720), *loxP-Piezo1* (029213), *Vglut2-ires2-Flpo* (030212), *inter-Ai65* (021875), *Vglut2-ires-cre* (016963), *Npy1r-gfp-cre* (030544), *P2ry1-ires-cre* (29284), *Pvalb-t2a-cre* (012358), *Crhr2-ires-Cre* (33728), *Npy2r-ires-Cre* (29285), *Calb1-ires2-cre* (28532), *Phox2b-cre* (16223), *Glp1r-ires-cre* (29283), *Mc4r-2a-Cre* (030759) and *Gpr65-ires-cre* (029282).

## Physiological measurements

Mice were anaesthetized with urethane ($1.6$–$1.8\,\mathrm{mg\,g^{-1}}$ intraperitoneal injection at least 30 min before surgery) or by isoflurane inhalation (1.5–2%) and warmed on a heated platform. Urethane was used for all experiments involving anaesthesia, except those in Extended Data Fig. 1, which explicitly describes the use of isoflurane in the figure legend. A tracheostomy was performed by inserting a cannula (18 or 20 gauge) to the carina and attaching the cannula to multipronged tubing with three openings: one to the atmosphere, one to a pressure transducer and one to an in-line gas and nebulized aerosol delivery port through which the animals are exposed to constant low-level flow rate ($40\,\mathrm{ml\,min^{-1}}$, which creates a tracheal pressure of $2$–$4\,\mathrm{mmH_2O}$, controlled by a SAR-1000 ventilator, CWE, room air in Fig. 1 and 100% oxygen in subsequent figures to minimize hypoxic sighs). The following parameters were measured: respiration was measured using an in-line pressure transducer; heart rhythm by electrocardiogram recorded with three needle electrodes placed subcutaneously in paws; oesophageal, pharyngeal and/or thoracic pressure by a fluid-filled pressure transducer; and respiratory muscle contraction by electromyographic recording with a concentric bipolar needle electrode coupled to an amplifier (1–2 kHz sampling, MP150 amplifier system, Biopac AcqKnowledge v.4.2, v.4.5 or v.5.0). Where indicated, electromyography

signals were digitally integrated ($\tau = 0.02\,\mathrm{s}$). Pulse oximetry monitoring was performed using a MouseSTAT Jr with a mouse paw sensor (Kent Scientific).

Thoracic compression was applied by affixing a cuff around the rib cage spanning from the forelimbs to the xiphoid process and inflating the cuff slowly over 5 s to achieve a 40–60% reduction in peak tracheal pressure per breath for 10 s, unless otherwise indicated. The cuff pressure varied by animal based on size and cuff fit, and the maximal pressure was typically $5$–$30\,\mathrm{cmH_2O}$. Airway suction was applied (5 s) by switching the in-line gas and nebulized aerosol delivery port to a digitally controlled vacuum reservoir (SCIREQ). The final applied suction pressure was determined by a pressure transducer in the trachea (low, $-5\,\mathrm{cm\,H_2O}$; high, $-10\,\mathrm{cm\,H_2O}$). Inhaled gases (Airgas, as in figures and legends, remaining percentage is $N_2$) were delivered in-line through the intake port on the ventilator ($40\,\mathrm{ml\,min^{-1}}$, 5 min trials). For measurements of the Hering–Breuer inspiratory reflex, lung inflation was achieved by increasing air flow through the ventilator ($10$–$25\,\mathrm{ml\,min^{-1}\,g^{-1}}$ body weight, 10 s). Aerosols were administered in saline (PBS) and delivered (5 s, room temperature) by a nebulizer (ANP-1100 from SCIREQ with a 50% duty cycle). Reflexes were monitored for the subsequent 5 min. Nebulized aerosols were methacholine ($10\,\mathrm{mg\,ml^{-1}}$, PBS, Cayman, 23092), citric acid (30% w/v, Sigma, C1909), KCl (Sigma, 12636) and microbeads (Thermo Scientific, 0.2 mm F8811, 2.0 mm F8827). Gasps were defined as single-breath expirations with >50% amplitude increase compared with the previous and subsequent breath, as inferred by electromyography and tracheal or oesophageal pressure measurements. For stimulus-evoked changes in breathing, data were normalized by comparison to values from a 10 s baseline immediately before stimulus introduction.

Respiratory mechanics (Extended Data Figs. 1e, 8c and 10f) were measured using a flexiVent computer-controlled piston ventilator (SCIREQ). Animals were anaesthetized, tracheostomized (18 or 20 g cannula inserted to the carina) and attached to the ventilator. In Extended Data Fig. 1e, closed-chest animals were then paralyzed ($1\,\mathrm{mg\,kg^{-1}}$ pancuronium, intraperitoneally, Sigma-Aldrich, P1918); measurements shown in Extended Data Figs. 8c and 10f were performed using open-chest animals. Mice were ventilated at 150 breaths per min, a tidal volume of $10\,\mathrm{ml\,kg^{-1}}$ and $3\,\mathrm{cmH_2O}$ positive end expiratory pressure with room air, unless otherwise indicated. Respiratory mechanics were assessed using the forced oscillation technique. Forced-expiratory volumes and pressure-volume loop manoeuvres were controlled by flexiVent software (flexiWare v.8.2).

## Calcium imaging in vagal ganglia

In vivo imaging of vagal ganglia was performed as previously described[20,56] with minor modifications. In brief, mice were anaesthetized with urethane as described above and given PBS (300 µl, intraperitoneally) early in the surgery for homeostatic support. The left vagal ganglia was surgically exposed with branches superior to the ganglion transected and immobilized on a glass imaging platform attached to a manipulator. Calcium imaging was performed in most experiments (4 out of 7 mice) by two-photon microscopy (Olympus FVMPE resonant-scanning two-photon microscope with a piezoelectric Z-stepper (P-915, Physik Instrumente) and ×25, NA1.0 water-immersion objective) using a Ti:sapphire laser with dispersion compensation (MaiTai eHP DeepSee, SpectraPhysics), with excitation tuned to 940 to 975 nm, and fluorescence emission filtered with a 570 nm long-pass dichroic and 495–540 nm bandpass filter for GCaMP6 and a 575–645 nm bandpass filter for TdTomato signals. Volumetric images were typically collected at 1.5–3 Hz with focal planes 40–60 µm apart (Olympus FluoView software vFV31S-SW). For some experiments (3 out of 7 mice), calcium imaging was performed by confocal microscopy (Leica SP5 II with ×20, NA1.0 water-immersion objective) as previously described[20].

Two-channel images were motion-corrected using the 'Image Stabilizer' plugin in Fiji ImageJ (v.1.52p). Red fluorescence channel images

were averaged to delineate individual cells and to demarcate regions of interest (ROIs). Unhealthy cells typically exhibited distinctively strong and unvarying GCaMP fluorescence relative to baseline and were excluded. Baseline fluorescence ($F_0$) was calculated from a 20 s period before stimulus onset, and ratiometric $\Delta F/F_0$ intensity was calculated and normalized to tdTomato fluorescence intensity at each ROI to control for photobleaching, motion and GCaMP6 expression. Cells were coded as responsive if stimulus-evoked increases in $\Delta F/F_0$ were at least 3 s.d. above the average fluorescence across the entire imaging session. For each responsive cell, the ratio ($R_c/R_i$) of response ($\Delta F/F_0$) to compression and inflation was calculated; cells were classified as compression-selective if $R_c/R_i > 2$, as inflation-selective if $R_c/R_i < 0.5$ or as polymodal if $0.5 < R_c/R_i < 2$. In Extended Data Fig. 2c, cells that did not respond to either airway inflation or airway closure were subsequently separated based on responsiveness to methacholine. Only some non-responsive neurons were selected for inclusion in indicated heatmaps based on computer randomization.

## Vagus nerve optogenetics
Vagus nerve optogenetics were performed as previously described[6,14] using a DPSS laser light source (473 nm, 150 mW, Ultralaser) with software actuated illumination (10 s, 5–40 Hz, 10 ms dwell, 65–95 mW mm$^{-2}$ Prizmatix Pulser v.2.3.1 TTL software).

## Cell ablations
Vagal sensory neurons were ablated as previously described[6,34] with DT (Sigma, D0564) solution (2–5 ng DT, PBS with 0.05% Fast Green FCF dye) injected (10 × 10 nl, serially) into surgically exposed vagal ganglia using a Nanoject III injector (Drummond). NEB ablation was achieved by intranasal administration (daily for 4 days) of solution containing 10 ng DT in 30 µl PBS. Cell ablation controls involved DT-administered Cre-negative littermates. Animals were allowed to recover for at least 2 weeks before subsequent experiments.

## Vagal ganglia injection
Vagal anatomical tracing was performed as previously described[6,14] and involved *AAV-eGFP* (AAV9.CB7.Cl.eGFP.WPRE.rBG, 105542-AAV9, Addgene) and *AAV-flex-TdTomato* (pAAV-FLEX-tdTomato, 28306-AAV9, Addgene). Animals recovered for at least 2 weeks before tissue collection.

## Whole-body plethysmography
Whole-body plethysmography was performed in freely behaving animals using a VivoFlow chamber system (SCIREQ). Chamber airflow was measured by a pneumotach at constant temperature and humidity with 0.5–0.6 l min$^{-1}$ bias flow, and respiratory measurements were amplified, digitized and recorded using the VivoFlow-usbAMP and Iox2 software (v.2.10.5.28, SCIREQ). Gas challenges involved hypoxia (12% $O_2$), hypercapnia (5% $CO_2$, 21% $O_2$) and normoxia (21% $O_2$) balanced with nitrogen (Airgas). Animals were acclimated in the plethysmography chamber for 40–60 min, and then baseline respiratory data were recorded for 30 min. CNO injections involved brief removal of the animal from the chamber for administration of CNO (3 mg kg$^{-1}$, intraperitoneally, 100 µl PBS), and animals were immediately returned to the chamber for further recordings (30 min). Breaths were assigned and respiratory parameters (tidal volume, breaths per minute (BPM), minute volume) were calculated using Iox2 software (v.2.10.5.28 SCIREQ). Gasp-like breaths were manually identified from pneumotachographs and defined as a 50% increase in both inspiration and expiration compared with preceding and subsequent breaths. For quantification of eupneic breathing parameters, data were filtered to exclude respiratory events outside typical adult mouse breathing (tidal volume >2 ml or <0.05 ml; BPM > 400), averaged over the recording period (with a 7 min delay after CNO introduction), and breathing measures dependent on airway volume were normalized to the body weight of the animal.

## Histology and expression analyses
For immunochemistry in tissue cryosections, tissues were collected from animals after transcardial perfusion of fixative (PBS followed by 4% paraformaldehyde in PBS), immersed in fixative (4% paraformaldehyde, PBS, overnight, 4 °C), cryopreserved (30% sucrose, PBS, overnight, 4 °C) and embedded in OCT. Tissue cryosections were obtained, washed (2 × 5 min, PBS, room temperature), permeabilized (0.3% Triton X-100, PBS, 10 min, room temperature), blocked (5% donkey serum, 0.3% Triton X-100, 0.05% Tween-20, PBS, 1 h, room temperature) and incubated with primary antibody diluted in blocking buffer (overnight, 4 °C; anti-NCAM1, 1:250, Cell Signaling Technology, 99746 S; anti-GFP, 5 µg ml$^{-1}$, Aves Labs, GFP-1020; anti-mCherry/RFP, 3 µg ml$^{-1}$, OriGene Technologies, AB0040-200; anti-HB-EGF (human), 1:250, R&D Systems, AF-259-NA; and anti-RFP, Rockland, 1:1,000, Rockland, 600-401-379). Slides were then washed (3 × 10 min, 0.3% Triton X-100, 0.05% Tween-20, PBS) and incubated with secondary antibodies in blocking buffer (4 h, room temperature, all 1:1,000, donkey polyclonal, Jackson Immunoresearch; anti-Chicken IgG-Alexa fluor 488, anti-rabbit IgG-Cy3, anti-rabbit IgG Cy5, anti-goat IgG Cy5 and anti-goat IgG Cy3; RRIDs: AB_2340375, AB_2307443, AB_2340607, AB_2340415 and AB_2307351, respectively). Samples were washed (3 × 10 min, 0.3% Triton X-100, PBS, room temperature), stained for nuclei visualization (5 min, 1:1,000 Hoechst 33342, PBS) and mounted (ProLong Glass Antifade; Thermo Fisher) for microscopy. RNA in situ hybridization for *Piezo2* was performed on tissue cryosections using the probe and protocol involving hybridization chain reaction provided by the manufacturer (Molecular Instruments). Immunostained slides and native tissue fluorescence were imaged by either confocal microscopy (Leica SP5 II or Nikon Ti2) or by widefield microscopy (Zeiss AxioZoom or AxioObserver microscopes with Zen Blue software, v.2.6 and v.3.2, respectively). For whole-mount lung histology in Fig. 3h, tissue was stained and cleared using published iDisco methodology involving anti-mCherry/RFP primary antibody (6 µg ml$^{-1}$) and Cy5-conjugated anti-goat IgG secondary antibody (1:500) and imaged by light sheet microscopy (UltraMicroscope II by LaVision, ImSpector v.7.1.4).

## Single-cell transcriptomics
Whole lungs below the trachea were collected from 10 *Calca-eGFP* and 10 *Ascl1-creER;lsl-tdTomato* mice (5–7 weeks old, equal male and female, 10 days after tamoxifen injection), pooled by strain, minced and incubated (60 min, 37 °C) in oxygenated papain dissociation buffer (Worthington Biochemical, LK003150). Residual tissue was mechanically dissociated through a 100 µm cell strainer, pelleted by centrifugation (400g, 7 min, 4 °C), washed, resuspended in red blood cell lysis buffer (150 mM $NH_4Cl$, 10 mM $NaCHO_3$ and 0.1 mM EDTA) for 5 min, pelleted and resuspended in FACS buffer (0.5% BSA, 2 mM EDTA, PBS, 4 °C). Immune cells were depleted using anti-CD45 magnetic beads according to the manufacturer's instructions (BioLegend, 480027), and the remaining cells were resuspended in viability buffer (TO-PRO-3 and CellTrace Violet, both 1:10,000, in RPMI 1640; Thermo Fisher, T3605, 65-0854-39 and 11835030, respectively). Cells were collected by FACS using a FACS Aria II (BD Bioscience) with gates to select for fluorescent reporter expression and viability (CellTrace Violet positive, TO-PRO-3 negative). Collected cells were individually encapsulated in nanodroplets using a 10x Genomics platform (v.3 chemistry). Single-cell cDNA was prepared according to the manufacturer's protocol and sequenced at the Harvard Medical School Biopolymers Facility on a NextSeq 500 platform. For analysis, sequence reads were aligned to the mm10 reference transcriptome, and feature barcode matrices were generated using Cell Ranger (10x Genomics; pipeline v.3.1.0) and analysed in R (v.4.1.3) using Seurat (v.4.1.1) for quality control, pre-processing, normalization, clustering and differential expression analysis. Transformed matrices from both strains were integrated (nFeature = 3,000) before cluster identification and UMAP representation. Analysis used

a standard process excluding cells with >15% mitochondrial reads or <500 unique features. Neuroendocrine cell clusters were identified for enriched expression of *Epcam*, *Calca* and *Ascl1*; genes to define additional lung cell types are depicted in Extended Data Fig. 9a. After differential expression analysis, gene ontology enrichment analysis used the top 50 most enriched genes ranked by significance (*P* value) using Enrichr[61] (https://maayanlab.cloud/Enrichr/).

## NEB calcium imaging

*Ascl1-creER;lsl-SALSA;lsl-Gα$_q$-DREADD* mice previously injected with tamoxifen (100 mg kg$^{-1}$ in sunflower oil, intraperitoneally, twice) were anaesthetized and transcardially perfused with 10 ml cold, oxygenated PBS. Lungs were inflated by introducing 2% low-melt agarose at 37 °C through a tracheal cannula and quickly chilled on ice (30 min). Lung lobes were resected, and 200-μm sections were obtained using a vibratome in cold, oxygenated imaging buffer (in mM: 115 NaCl, 5 KCl, 25 NaHCO$_3$, 1 MgCl$_2$, 2 CaCl$_2$, 10 glucose and 10 HEPES, pH 7.3). Slices were transferred to fresh imaging buffer (37 °C; 5% CO$_2$) for imaging (typically 30 min later). NEBs were identified based on tdTomato expression, and SALSA fluorescence was measured by confocal microscopy (Leica SP5 II with ×20, NA1.0 water-immersion objective, GCaMP6f, 488 nm excitation and 495–535 emission; tdTomato, 543 nm excitation and 565–615 emission). Imaging was performed with continuous perfusion of imaging buffer by gravity feed, and application of CNO or KCl as indicated in Extended Data Fig. 6a. Data were acquired using LAS AF software (v.2.3.6 Leica) and analysed in ImageJ.

## Whole-nerve electrophysiology

Whole vagus nerve electrophysiology recording was performed as previously described[14,21,53] with minor modifications. In brief, urethane-anaesthetized animals (1.6 mg g$^{-1}$) were surgically prepared to administer airway suction, as described above for airway physiology measurements. The left vagus nerve was then transected, and the lung-connected nerve end was desheathed and placed on a pair of platinum–iridium electrodes. The nerve and electrode were immersed in halocarbon oil, and a ground electrode was placed on nearby muscle. Multiunit nerve activity was amplified (CP511, Grass Technologies), digitized (MP150, Biopac), recorded (AcqKnowledge software, v.4.5, Biopac) and integrated (Elenco, RS-400). Stimulus-induced responses were calculated as a percentage change from baseline activity and normalized to the response to serotonin (intraperitoneally, 10 mM, 400 μl PBS) over 100 s after administration.

## Behaviour coding

Animals were video recorded (Logitech C920 HD PRO camera) during whole-body plethysmography. After acclimation (1 h), behaviours were manually scored using BORIS software (v.8.20.4)[62] by a genotype-blinded investigator who measured time exploring, rearing, grooming, sniffing or hunching for 10 min periods before and at minutes 7–17 after CNO administration (3 mg kg$^{-1}$, intraperitoneally). Hunching was defined based on a characteristic recumbent posture and was typically associated with laboured breathing and ruffled fur.

## Statistics and reproducibility

Data in graphs are presented as the mean ± s.e.m., unless otherwise indicated. Statistical analyses were performed using Prism (GraphPad) with statistical tests and sample sizes reported in figures and legends. All replicates were biological, unless otherwise indicated, and statistical tests were two-sided. All representative images are from at least three independent experiments. Sample sizes were determined based on previous expertise and publications in our field. Investigators were blinded to group allocations for plethysmography, physiological and behavioural experiments associated with Figs. 3–5 and Extended Data Fig. 7; group allocation was not blinded in other experiments. Where appropriate, exact and adjusted *P* values are reported in legends, and asterisks for significance are defined as follows: *$P < 0.05$, **$P < 0.01$, ***$P < 0.001$, ****$P < 0.0001$.

## Materials availability

All reagents that are not commercially available will be made available upon reasonable request. *Olfr78-p2a-cre* mice will be deposited to the Jackson Laboratory and will be available following completion of a standard material transfer agreement.

## Reporting summary

Further information on research design is available in the Nature Portfolio Reporting Summary linked to this article.

## Data availability

Single-cell transcriptome data for NEBs are publicly available at the NCBI's Gene Expression Omnibus (GSE252735). Source data are provided with this paper.

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

**Acknowledgements** We thank M. Allison, S. Prescott and C. Kelsey for comments on the manuscript; and staff at the Harvard Medical School Nikon Imaging Facility, Neurobiology Imaging Facility, Biopolymers Facility, and Immunology Flow Cytometry Core for technical assistance. This work was supported by grants from the NIH (DP1 AT009497 and R01 HL132255 to S.D.L. and F32 HL156583 to M.S.S.) and the Food Allergy Science Initiative to S.D.L. R.S.G. is supported by a Damon Runyon fellowship, A.A.A. is supported by a John S. LaDue Fellowship, and S.D.L. is an investigator of the Howard Hughes Medical Institute.

**Author contributions** M.S.S. and S.D.L. designed experiments. M.S.S., N.R.J., R.S.G., P.A.B., S.M., A.A.A. and C.Z. performed experiments. M.S.S., P.A.B. and S.D.L. analysed data. M.S.S. and S.D.L. wrote the manuscript.

**Competing interests** S.D.L. is a consultant for Kallyope. The remaining authors declare no competing interests.

### Additional information

**Correspondence and requests for materials** should be addressed to Stephen D. Liberles.

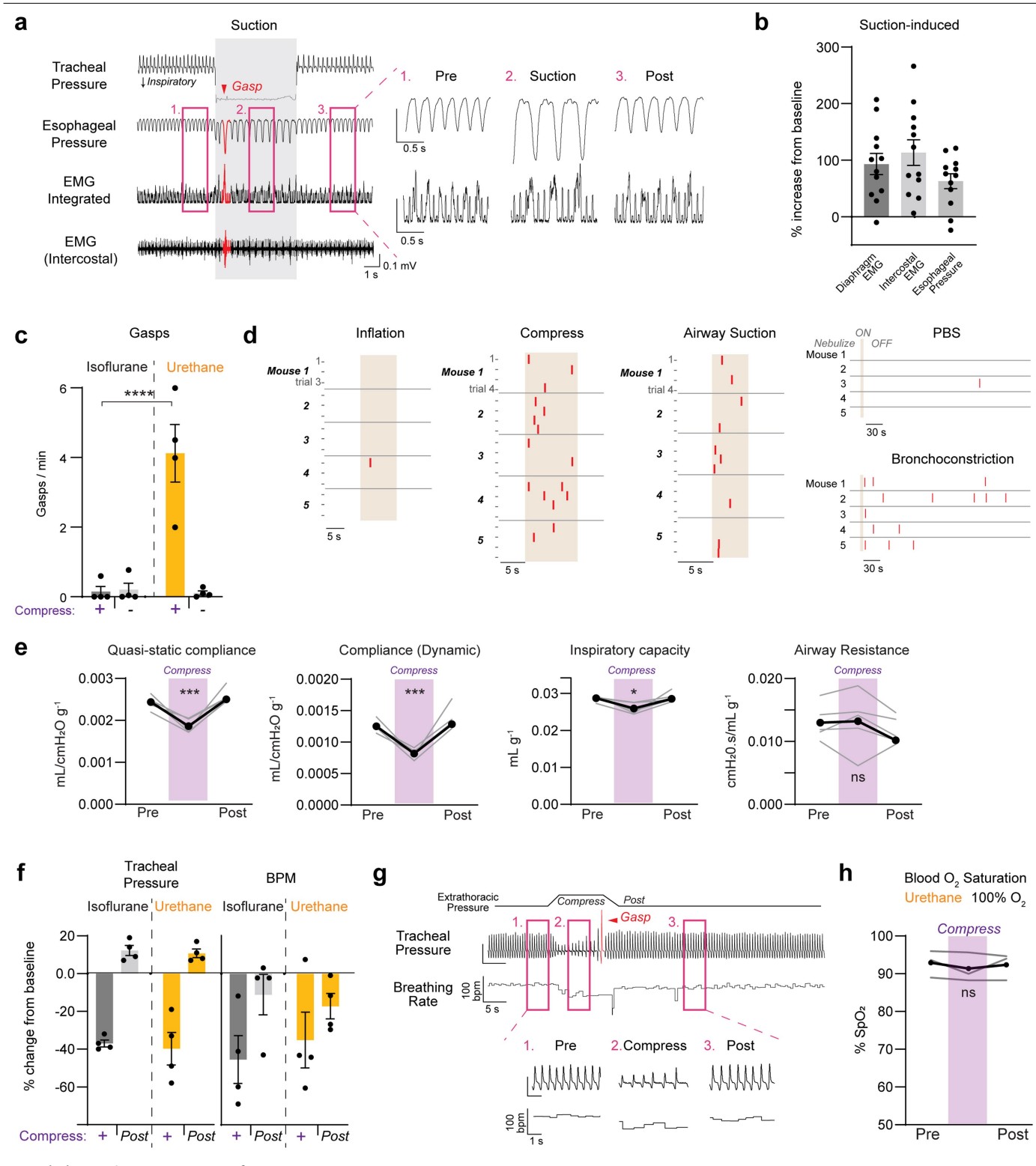

**Extended Data Fig. 1** | See next page for caption.

**Extended Data Fig. 1 | Respiratory measurements of airway closure.**
**a**, Representative physiological measurements before, during and after suction (grey shading) with magnifications (right) depicting data from magenta boxes, red: gasps. **b**, Quantification of physiological changes induced by airway suction (low), expressed as a percentage increase per breath from baseline (n = 12 trials from 3 animals), mean ± sem, dots: individual trials, one-way ANOVA with Bonferroni post hoc test: Treatment x Measure $F_{(1.57, 17.32)}$ = 2.753, ns. **c**, Quantification of gasp frequency with or without thoracic compression, mean ± sem, dots: average per animal across at least three trials, n = 4 animals per group, two-way ANOVA with Bonferroni post hoc test: Anesthesia x Condition $F_{(1,8)}$ = 37.19, p = 0.0003; multiple comparisons (Isoflurane versus Urethane): (−), ns; (+), p < 0.0001. **d**, Raster plots indicating gasps (red stripe) before, during (beige shading), and after stimuli indicated, 1-4 trials per each of 5 mice. **e**, Respiratory measurements before, during and after thoracic compression (purple shading) measured in ventilated mice, black: mean, grey: individual animals, (n = 5 mice, one-way ANOVA with Bonferroni post hoc test: Inspiratory Capacity, $F_{(1.2, 4.9)}$ = 10.12, p = 0.0229; Airway Resistance, $F_{(1.1, 4.5)}$ = 1.21, p = 0.3377; Quasi-static Compliance, $F_{(1.5, 6.0)}$ = 31.02, p = 0.0009; Dynamic compliance, $F_{(1.4, 5.6)}$ = 35.01, p = 0.0009). **f**, Quantification of physiological measurements during (+) and 10 sec following thoracic compression (Post) under isoflurane (grey) or urethane (yellow) anesthesia in freely breathing mice, mean ± sem, dots: average per animal across at least three trials, n = 4 mice per group, two-way ANOVA with Bonferroni post hoc test (Isoflurane versus Urethane, pairwise comparisons non-significant). **g**, Representative physiological measurements before, during and after airway compression magnifications below depicting data from magenta boxes, red arrow: gasps. **h**, Pulse oximetry measurements before, during and after thoracic compression (purple shading) measured in freely-breathing urethane anesthetized mice, black: mean, grey: individual animals, n = 3 mice, one-way ANOVA with Bonferroni post hoc test: Condition $F_{(1.01, 2.02)}$ = 1.010, p = 0.4211, all pairwise comparisons non-significant; ns: non-significant.

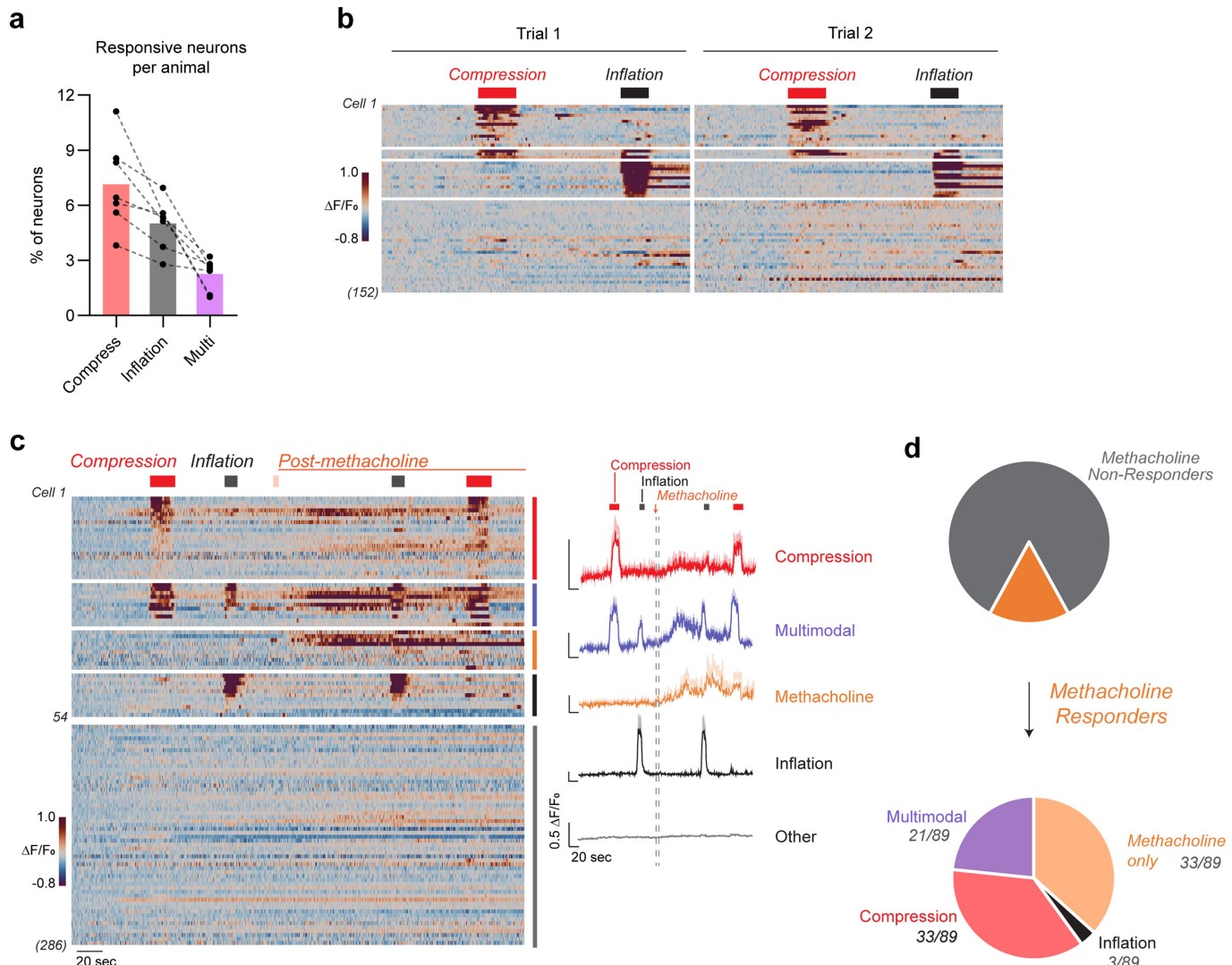

**Extended Data Fig. 2 | Imaging vagal responses to airway stimuli.**
**a**, Quantification of neurons responsive to compression only, inflation only, or both, data from individual mice (dots, n = 7) are connected (dashed lines), bar: mean. **b**, Heat map depicting calcium responses in vagal sensory neurons (ΔF/F color coded) to airway compression (red bar) and airway inflation (black bar) across multiple trials; 152 neurons were imaged, all responsive and only some randomly selected non-responsive neurons are shown. **c**, Heat map (left) depicting vagal sensory neuron calcium responses (ΔF/F color coded) to airway compression (red bar), airway inflation (black bar), and nebulized methacholine (orange bar); 286 neurons were imaged, all responsive and only some randomly selected non-responsive neurons are shown. Average neuron response (right) of indicated type to stimuli depicted; dark line: mean, light line: SEM, dotted line: time of methacholine administration. **d**, Proportion of neurons responsive to nebulized methacholine (orange slice, 89/468 neurons, 2 mice) and distribution of methacholine-responsive neurons activated by compression and/or inflation.

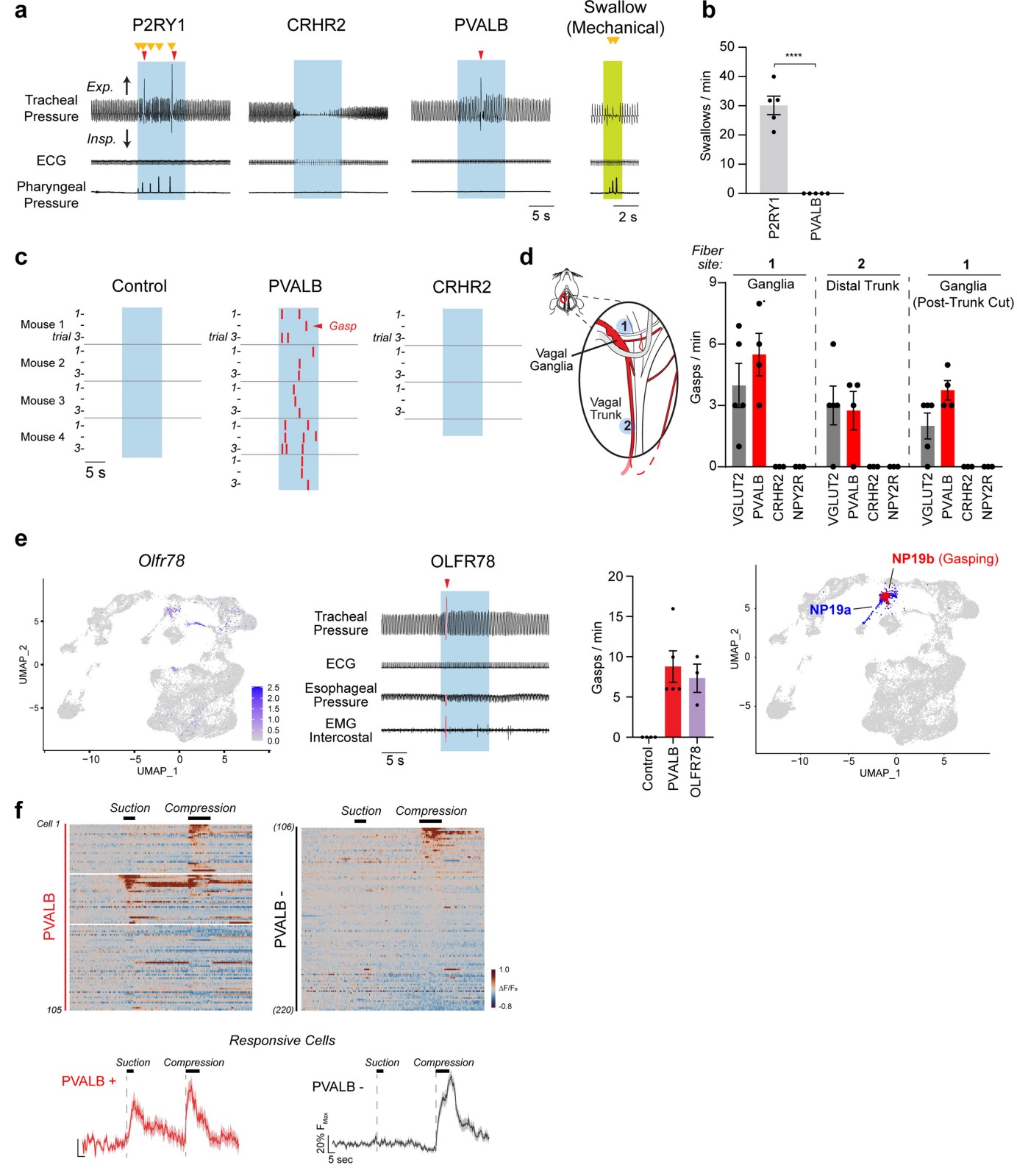

**Extended Data Fig. 3** | See next page for caption.

**Extended Data Fig. 3 | Physiological responses to optogenetic stimulation of various vagal sensory neuron types. a**, Representative physiological measurements before, during and after optogenetic stimulation (blue shading, ganglion illumination after cutting of vagal trunk) of vagal P2RY1, CRHR2, and PVALB neurons or, for comparison, after mechanically-evoked swallow (green shading), Exp: expiratory phase; Insp: inspiratory phase, red triangle: gasp, yellow triangle: swallow. **b**, Quantification of swallows evoked from optogenetic stimulation of indicated vagal neurons, mean ± sem, dots: average per animal across at least three trials, n = 5 mice per group; unpaired t test, p < 0.0001. **c**, Raster plots indicating gasps (red stripe) before, during (blue shading), and after optogenetic stimulation of vagal neurons indicated, 3 trials per each of 3-5 mice. **d**, Quantification of gasp frequency (right) to optogenetic stimulation of vagal ganglia and trunk as numbered in the cartoon (left), mean ± sem, dots: average per animal across at least three trials, n = 5 VGLUT2, 4 PVALB, 3 CRHR2, 3 NPY2R. **e**, (left) UMAP plot of *Olfr78* expression in vagal sensory neurons based on published transcriptomic data[6]; (center-left) Representative physiological measurements before, during and after optogenetic stimulation as described in **a**; (center-right) Quantification of gasps evoked in indicated animals, PVALB data reproduced from Fig. 3b, mean ± sem, dots: average per animal across at least three trials, n = 4 Control, 5 PVALB, 3 OLFR78 mice; (right) UMAP plot indicating subtypes of NP19 neurons that express *Pvalb* and mediate gasping (red dots) or do not (blue dots). **f**, Heat map (top) depicting vagal sensory neuron calcium responses ($\Delta F/F$ color coded) in *Snap25-GCamp6s; lsl-TdTomato; Pvalb-t2a-cre* mice to airway suction (high) and compression in tdTomato-positive (left) or tdTomato-negative neurons, 105 tdTomato-positive and 390 tdTomato-negative neurons were imaged, all responsive and only some randomly selected non-responsive neurons are shown; average $\Delta F/F$ traces of all responsive neurons (bottom, 57 PVALB-positive, 46 PVALB-negative neurons) to indicated stimuli, dark line: mean, light line: SEM, dotted line: stimulus onset.

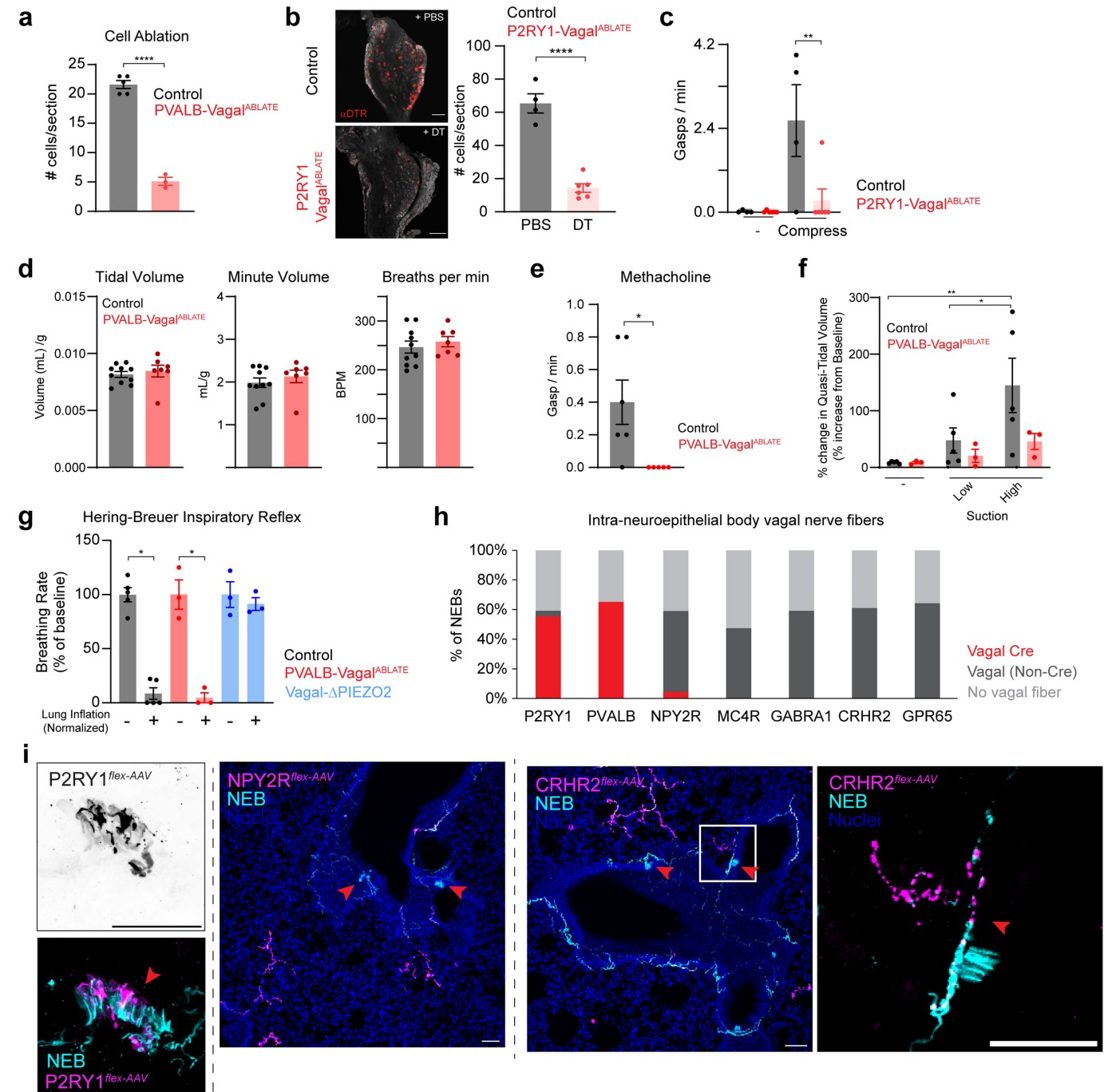

**Extended Data Fig. 4** | See next page for caption.

**Extended Data Fig. 4 | Manipulation and visualization of vagal neuron types. a**, Quantification of PVALB neurons in vagal ganglia of mice indicated, mean ± sem, dots: individual animals with cell counts averaged across at least 3 cryosections per animal, n = 5 Control, 3 PVALB-Vagal$^{ABLATE}$ mice; unpaired t test, p < 0.0001. **b**, Representative image (left) of DTR immunostaining in vagal ganglia of *P2ry1-ires-Cre; lsl-DTR* mice after vagal injection with PBS (Control) or DT (P2RY1-Vagal$^{ABLATE}$), scale bar: 100 μm. Quantification of P2RY1 vagal neuron ablation (right), mean ± sem, dots: individual animals with cell counts averaged across at least 3 cryosections per animal, n = 4 Control, 6 P2RY1-Vagal$^{ABLATE-}$ mice; unpaired t test, p < 0.0001. **c**, Quantification of gasp frequency in response to thoracic compression in P2RY1-Vagal$^{ABLATE}$ mice or control littermate mice from **b**, n = 4 Control, 6 P2RY1-Vagal$^{ABLATE-}$ mice; mean ± sem, dots: average responses of individual animals across at least 3 trials per mouse, two-way ANOVA with Bonferroni post hoc test: Condition x Ablation $F_{(1,8)}$ = 6.319, p = 0.0362; Control versus P2RY1-Vagal$^{ABLATE}$: (−), non-significant; Compress, p = 0.0054. **d**, Quantification of breathing measurements by whole body plethysmography in mice described in panel **a**; mean ± sem, dots: individual animals (n = 10 Control, 7 PVALB-Vagal$^{ABLATE}$, unpaired t test: all ns). **e**, Quantification of gasps following nebulized methacholine delivery in urethane-anesthetized mice, mean ± sem, dots: individual animals, n = 6 Control, 5 PVALB-Vagal$^{ABLATE}$; unpaired t-test: p = 0.0266. **f**, Quantification of quasi-tidal volume measured by esophageal pressure at baseline (-), low suction (-5 cmH$_2$O) or high airway suction (-10 cmH$_2$O) in urethane-anesthetized PVALB-Vagal$^{ABLATE}$ mice or control DT-injected littermates lacking Cre; mean ± sem, dots: average responses of individual animals across 4 trials per mouse, n: 5 Control, 3 PVALB-Vagal$^{ABLATE}$ mice; two-way ANOVA with Bonferroni post hoc test; Condition x Genotype $F_{(2,12)}$ = 2.088, p = 0.1667; pairwise comparison (−) versus High suction (Control), p = 0.0024; low suction versus high suction (Control), p = 0.0240; other pairwise comparisons, ns. **g**, Assessment of the Hering-Breuer inspiratory reflex by measuring breathing rate at increased lung volumes (air flow/g body weight); mean ± sem, n = 5 control, 3 PVALB-Vagal$^{ABLATE}$, 3 Vagal-ΔPiezo2 (*Phox2b-cre; loxP-Piezo2*) animals; two-way ANOVA with Bonferroni post hoc test: Genotype, $F_{(2,8)}$ = 25.75, p = 0.0003,; pairwise comparisons (Baseline versus Inflation, left to right): p = 0.0001, 0.0005, ns. **h**, Quantification of NEBs (visualized by NCAM1 staining) co-localized with nerve fibers labeled by immunohistochemistry for tdTomato and GFP after injection of AAVs containing a Cre-dependent *tdTomato* allele and a Cre-independent *Gfp* allele into vagal ganglia of Cre lines indicated. NEBs analyzed (left-to-right): 27, 25, 22, 21, 27, 36, and 14 from at least 3 lung sections derived from at least 2 mice per group. **i**, Representative images of cryosections from **h** depicting immunohistochemistry for TdTomato (magenta) and anti-NCAM1 (cyan); CRHR2 depicts alternate wideview of Fig. 3i image and additional ROI (white box); red arrows: NEBs, scale bars: 50 μm.

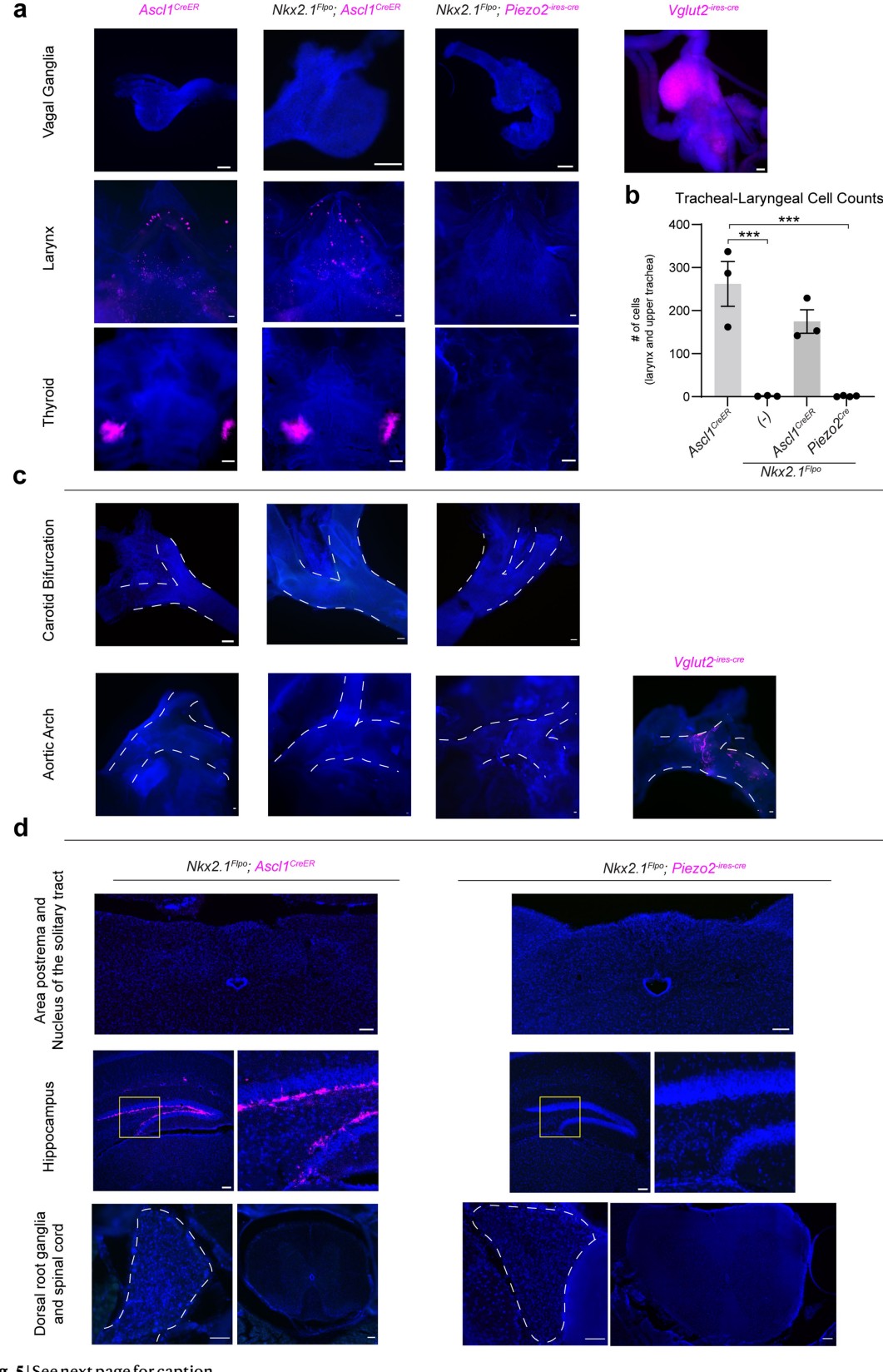

**Extended Data Fig. 5** | See next page for caption.

**Extended Data Fig. 5 | Validating intersectional genetic tools to access NEBs. a**, Representative widefield images of native reporter fluorescence (magenta) of wholemount tissue from *Ascl1-CreER;; lsl-TdTomato* mice (left*), Ascl1-CreER; Nkx2.1-Flpo: inter-Gα$_q$-DREADD-mCherry*, mice (second from left), *NEB$^{Gαq-DREADD}$-mCherry* mice (third from left), and *Vglut2-ires-cre; lsl-TdTomato* mice (right), images representative of tissue collected from at least 4 animals per genotype, scale bars: 100 μm (vagal ganglia and larynx) and 500 μm (thyroid). **b**, Quantification of tracheal-laryngeal cells labeled by indicated mouse lines described in **a;** mean ± sem, dots: individual animal, n = 3 per indicated genotype, 4 *NEB$^{Gαq-DREADD}$-mCherry*; one-way ANOVA with Bonferroni post hoc test: $F_{(3,9)}$ = 23.63, p = 0.0001, pairwise comparisons (left to right versus Ascl1-CreER): p = 0.0003; ns, p = 0.0002.**c**, Representative widefield images of native reporter fluorescence (magenta) in indicated tissues from mice described in **a**. Dotted line indicates blood vessel, scale bar: 100 μm. **d**, Representative widefield images of native reporter fluorescence (magenta) in indicated tissues from mice described in **a**. In middle panels, the yellow box depicts site of image on right. In lower panels, the dotted line demarks the dorsal root ganglia (left), spinal cord (right), scale bars: 100 μm.

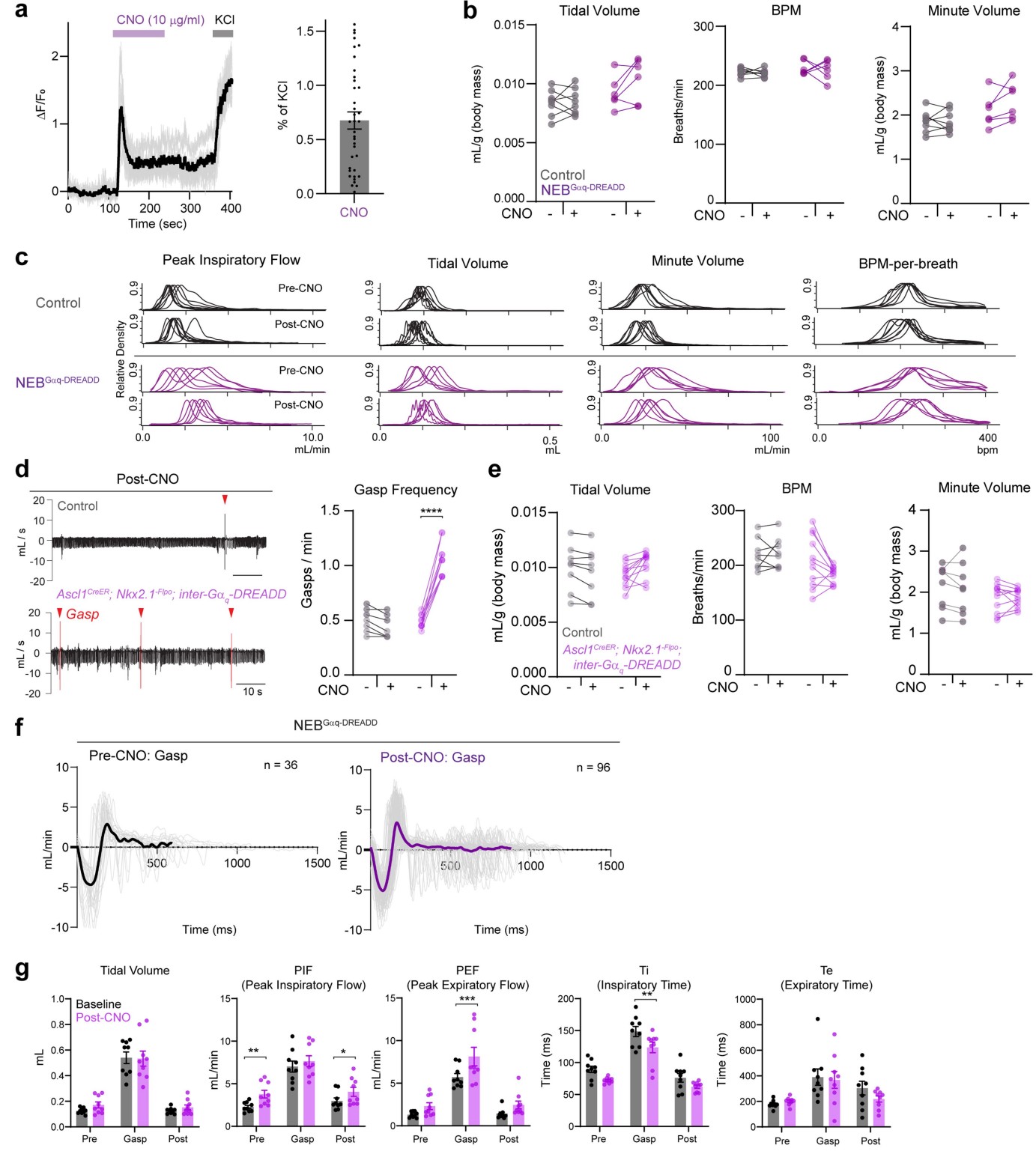

**Extended Data Fig. 6** | See next page for caption.

**Extended Data Fig. 6 | Breathing changes following NEB activation. a**, (left) Calcium responses ($\Delta F/F_0$) of individual NEB cells in ex vivo lung slices from *Ascl1-CreER; lsl-SALSA; lsl-G$\alpha_q$-DREADD* mice, dark line: mean, grey lines: individual cells, n = 12 cells from 3 mice; (right) Quantification of CNO-evoked calcium responses normalized to KCl (40 mM) responses, mean ± sem, dots: individual cells, n = 37 cells from 3 mice. **b**, Quantification of whole body plethysmography measurements in *NEB$^{G\alpha q\text{-}DREADD}$* mice and littermate controls lacking Cre before and for 10 min after CNO administration, dots: individual animals, lines: differences per animal across conditions, n = 8 control, 6 *NEB$^{G\alpha q\text{-}DREADD}$* mice; two-way ANOVA with Bonferroni post hoc test: Condition (CNO) x Genotype (Tidal Volume, BPM (breaths per minute), and Minute Volume, respectively) $F_{(1,12)}$ = 2.498, 0.010, 2.531; p value all non-significant (ns). **c**, Histogram of individual breath measurements before and after CNO administration; lines: distribution of per breath measurements across one animal, n = 7 Control, 6 *NEB$^{G\alpha q\text{-}DREADD}$*. **d**, Pneumotachographs of airflow in freely behaving *Ascl1-CreER; Nkx2.1-Flpo; inter-G$\alpha_q$-DREADD* mice (left) before and 10 min after CNO administration (3 mg/kg, IP), red arrows: gasps. Quantification of gasp frequency (right) in *Ascl1-CreER; Nkx2.1-Flpo; inter-G$\alpha_q$-DREADD* mice and littermate controls before and over 20 min after CNO administration, dots:

individual animals, lines: differences per animal across conditions, n = 8 control, 11 *Ascl1-CreER; Nkx2.1-Flpo; inter-G$\alpha_q$-DREADD* mice; two-way ANOVA with Bonferroni post hoc test: Condition x Genotype $F_{(1,17)}$ = 144.1, p < 0.0001; pairwise test (Pre-CNO versus Post-CNO): control, ns; *Ascl1-CreER; Nkx2.1-Flpo; inter-G$\alpha_q$-DREADD* mice, p < 0.0001. **e**, Quantification of breathing parameters in mice from **d**, two-way ANOVA with Bonferroni post hoc test: Condition x Genotype (Tidal Volume, BPM, and Minute Volume, respectively) $F_{(1,17)}$ = 10.96, 3.066, 0.2292; p = 0.0041, ns, ns; pairwise comparisons for Tidal Volume: Control, ns; *Ascl1-CreER; Nkx2.1-Flpo; inter-G$\alpha_q$-DREADD* mice, p = 0.0038. **f**, Mean pneumotachographs of gasps in *NEB$^{G\alpha q\text{-}DREADD}$* mice before CNO (left) and 10 min after CNO (right) during whole body plethysmography; dark trace: mean, grey trace: individual breaths (n = 36 breaths pre-CNO, 96 breaths post-CNO from 5 mice). **g**, Quantifying kinematics of spontaneous gasps and CNO-induced gasps in *NEB$^{G\alpha q\text{-}DREADD}$* mice, as well as the preceding and subsequent breaths; mean ± sem, dots: individual breaths (n = 9 paired breaths in each condition from 3 mice; two-way ANOVA with Bonferroni post hoc test: Time (Pre, Gasp, Post) x Treatment (CNO): $F_{(2,16)}$=ns for all measures; significant pairwise tests (Baseline versus Post-CNO), left to right: p = 0.0020, 0.0259, 0.0001, 0.0058).

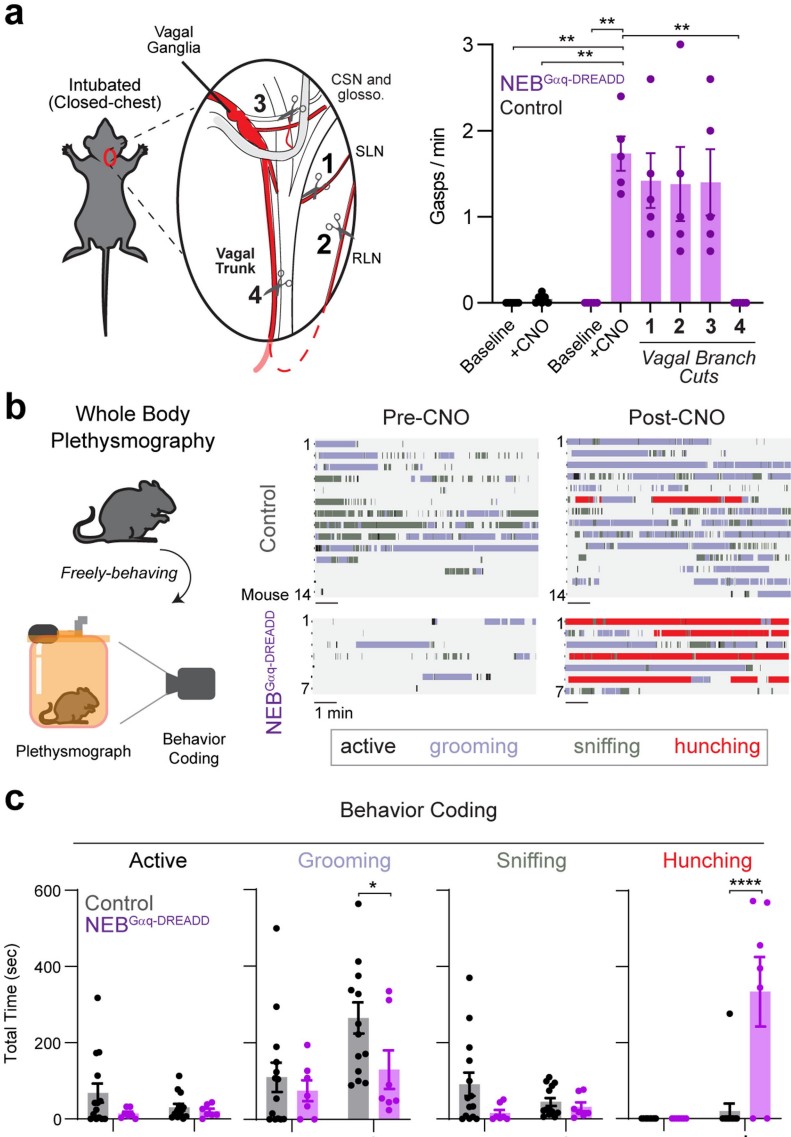

**Extended Data Fig. 7 | Behavioral and physiological responses to NEB activation. a**, Quantification (right) of gasp frequency in urethane-anesthetized *NEB^{Gαq-DREADD}* mice and littermate controls before and after CNO (IP) and serial transection of nerve branches as numbered in the cartoon (left); mean ± sem, dots: individual animals, n = 6 Control, 5 *NEB^{Gαq-DREADD}* mice; one-way ANOVA with Bonferroni post hoc test: $F_{(1.18,4.86)} = 17.98$; pairwise comparisons to *NEB^{Gαq-DREADD}* (+CNO): Control (baseline), p = 0.0058; Control (+CNO), p = 0.0072; *NEB^{Gαq-DREADD}* (baseline), p = 0.0069; Cut 4, p = 0.0069; all other

comparisons, ns). **b**, (left) Cartoon of plethysmography with simultaneous video recording; (right) Ethograms from *NEB^{Gαq-DREADD}* mice and littermate controls before and after CNO administration, n = 14 control, 7 *NEB^{Gαq-DREADD}* mice; row: individual animal. **c**, Quantification of behavior from animals in **b**, dots: individual animals, bars: mean ± sem, two-way ANOVA: Treatment x Genotype (Active, Grooming, Sniffing, Hunching, respectively) $F_{(1,19)}$ = ns, ns, ns, 20.61; pairwise comparisons (Control vs. *NEB^{Gαq-DREADD}*) for Grooming, p = 0.0392; Hunching, p < 0.0001.

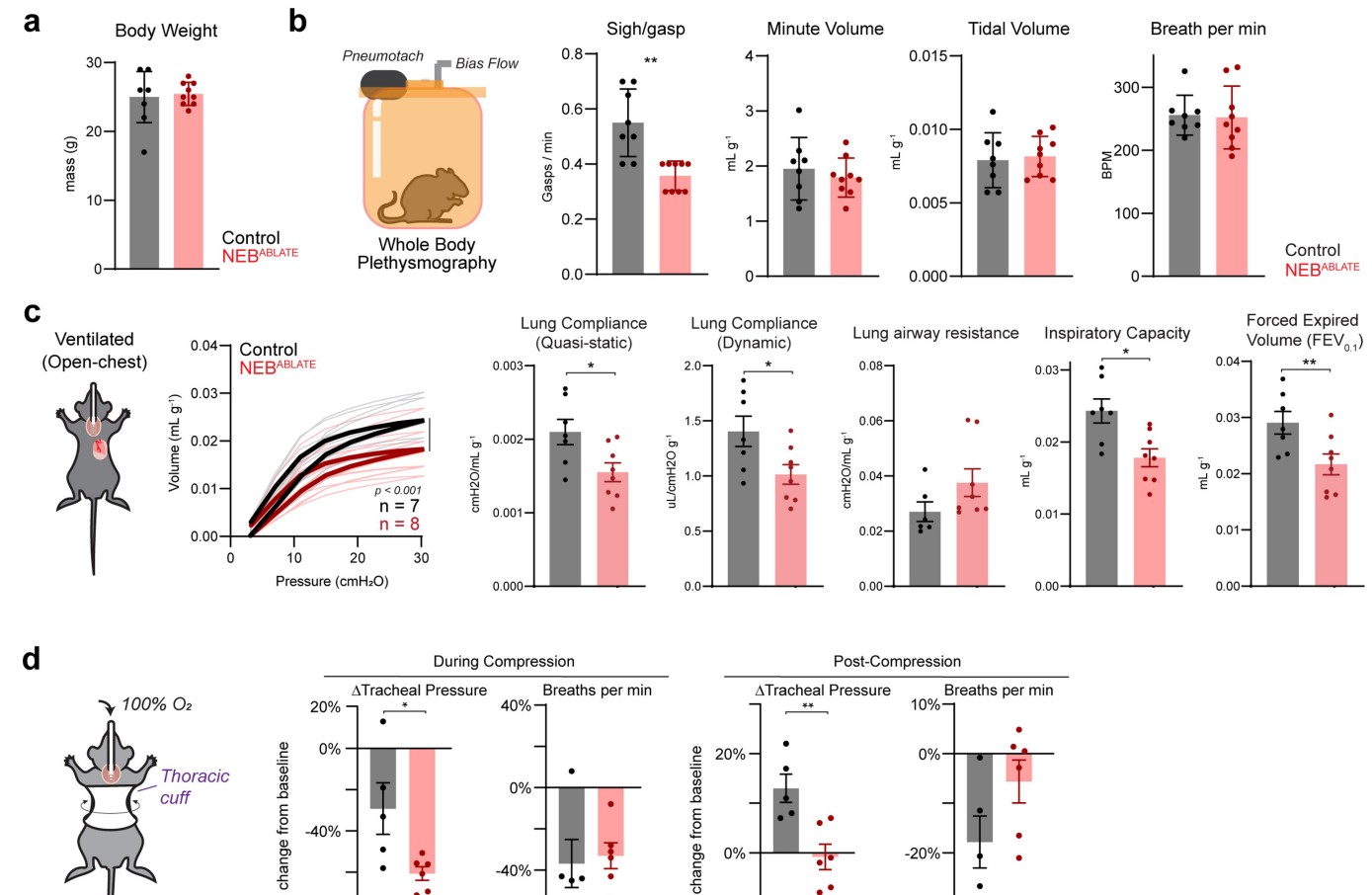

**Extended Data Fig. 8 | Physiological measurements after NEB ablation.**
**a**, Quantification of body weight in NEB$^{ABLATE}$ (red) and littermate control mice without Cre expression (gray); mean ± sem, dots: individual animals, n = 7 Control, 9 NEB$^{ABLATE}$; unpaired t test, ns. **b**, Quantification (right) of breathing measurements by whole body plethysmography (cartoon, left) in mice in NEB$^{ABLATE}$ (red) and littermate controls (gray); mean ± sem, dots: individual animals, n = 8 Control, 9 NEB$^{ABLATE}$ mean ± sem, unpaired t test (left-to-right): p = 0.002, ns, ns, ns. **c**, Quantification of respiratory mechanics (right) and pressure-volume loop test (middle) performed in ventilated (cartoon, left

control and NEB$^{ABLATE}$ mice; n (left to right) = 7, 7, 6, 7, 7 (Control), all 8 (NEB$^{ABLATE}$) mice, (middle) thick line: mean, thin lines: individual animals, Mann-Whitney test: p < 0.0001; (right) mean ± sem, dots: individual animals, unpaired t test (left-to-right): p = 0.0218, 0.029, ns, 0.0073, 0.0182. **d**, Quantification of physiological measurements during and post (10 sec following) thoracic compression (cartoon, left) in freely breathing mice; n = 5 Control, 6 NEB$^{ABLATE}$ mice; mean ± sem, dots: average of three trials per animal, unpaired t test, left-to-right: p = 0.0265, ns, 0.0057, 0.1029.

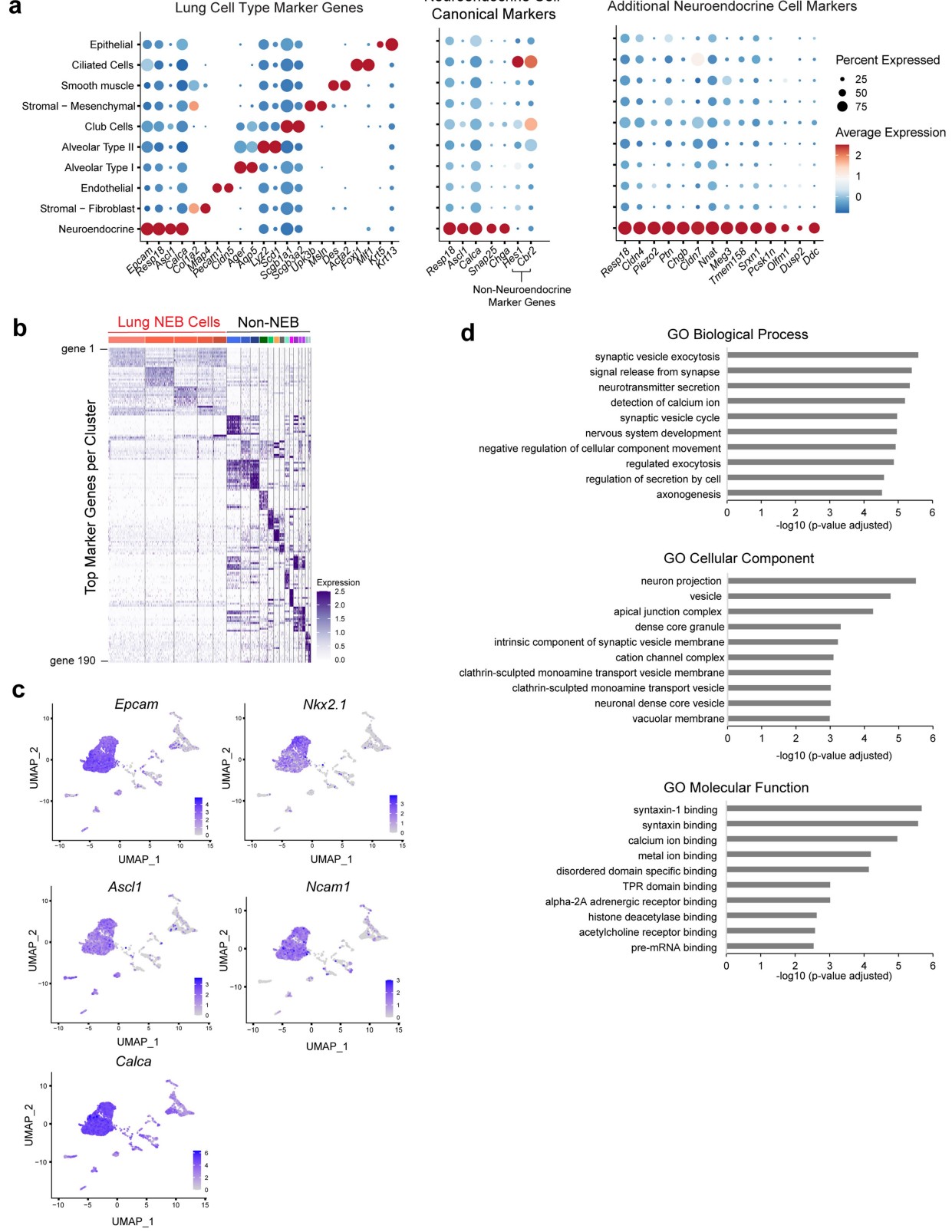

**Extended Data Fig. 9 | Single-cell transcriptomes of NEBs. a**, Dot plots of lung cell type marker genes and neuroendocrine cell-enriched genes reported previously[44,52,63,64] or identified here by differential expression analysis. **b**, Heat map of top differentially expressed genes across cell types in Fig. 5b. **c**, UMAP of lung epithelial and NEB-enriched genes (scale: natural log). **d**, Gene ontology (GO) terms of top 50 genes enriched in NEBs; adjusted p-values are computed by the Benjamini-Hochberg method for correction for multiple hypothesis testing using Enrichr.

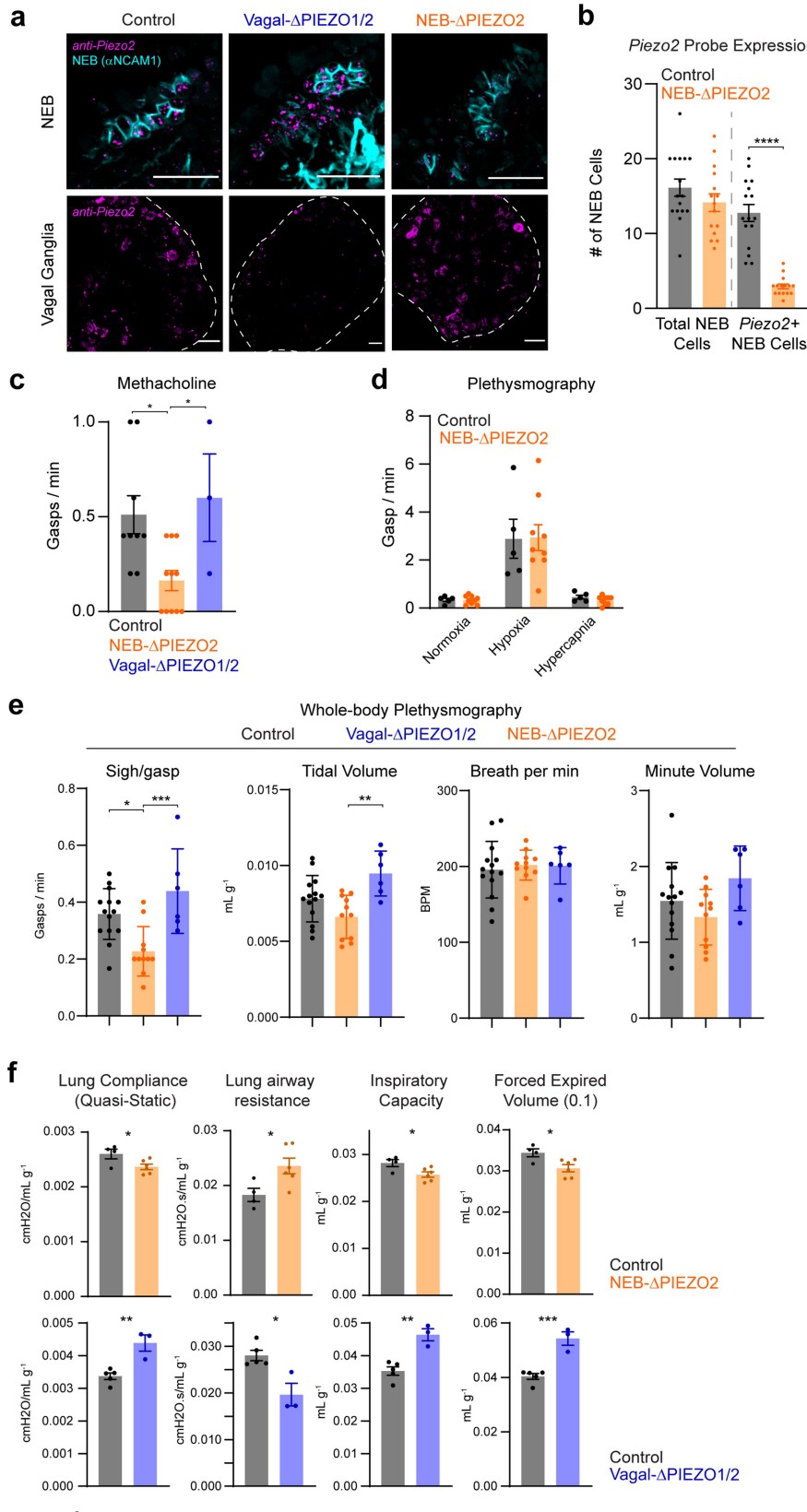

**Extended Data Fig. 10** | See next page for caption.

**Extended Data Fig. 10 | Breathing changes in cell-specific PIEZO knockout mice. a**, Representative images of *Piezo2* RNA in situ hybridization (magenta) in NEBs and vagal ganglia, NEBs visualized by NCAM immunochemistry (cyan), scale bar: 50 μm. **b**, Number of total NEB cells (NCAM1 immunoreactivity) and *Piezo2*-expressing NEB cells from mice in **a**, mean ± sem, dots: individual sections, n = 16 sections from 2 Control mice, 15 sections from 3 NEB-ΔPIEZO2 mice, one-way ANOVA with Bonferroni post hoc test: *Piezo2* NEB-ΔPIEZO2 vs. all comparisons, p < 0.0001; all other comparisons, ns. **c**, Quantification of gasps following nebulized methacholine delivery in urethane-anesthetized mice, mean ± sem, dots: individual animals, n = 9 Control, 11 NEB-ΔPIEZO2, 3 Vagal-ΔPIEZO1/2, one-way ANOVA with Tukey post hoc test: $F_{(2,20)}$ = 5.923, p = 0.0095; pairwise comparisons: Control versus NEB-ΔPIEZO2, p = 0.0198; Vagal-ΔPIEZO1/2 versus NEB-ΔPIEZO2, p = 0.0457; all other pairwise comparisons, ns. **d**, Quantification of gasps by whole body plethysmography during normoxia (21% $O_2$), hypoxia (12% $O_2$), and hypercapnia (5% $CO_2$) (7 min), mean ± sem, dots: individual animals, n = 5 control, 9 NEB-ΔPIEZO2, two-way ANOVA with Bonferroni post hoc test: Condition x Genotype $F_{(2,24)}$ = 0.02893, p = 0.9715; Control versus NEB-ΔPIEZO2, all pairwise comparisons, ns. **e**, Quantification (right) of breathing measurements made by whole body plethysmography (cartoon, left), mean ± sem, dots: individual animals, n = 14 Control, 11 NEB-ΔPIEZO2, 6 Vagal-ΔPIEZO1/2, one-way ANOVA with Bonferroni post hoc test: Gasps, $F_{(2,28)}$ = 9.9558, p = 0.0007, Control versus NEB-ΔPIEZO2, p = 0.0104, Vagal-ΔPIEZO1/2 versus NEB-ΔPIEZO2, p = 0.0010; Tidal Volume, $F_{(2,28)}$ = 7.263, p = 0.0029, Vagal-ΔPIEZO1/2 versus NEB-ΔPIEZO2, p = 0.002; all other ANOVA results and pairwise comparisons, ns. **f**, Quantification of respiratory mechanics performed in NEB-ΔPIEZO2 (top, orange, n = 6), Vagal-ΔPIEZO1/2 (bottom, blue, n = 3) mice and control littermates without Cre expression (grey, n = 4 left, 5 right), mean ± sem, dots: individual animals, unpaired t test, NEB-ΔPIEZO2 left-to-right: p = 0.0384, p = 0.0314, p = 0.0285, p = 0.0223; Vagal-ΔPIEZO1/2: p = 0.0040, p = 0.0106, p = 0.0022, p = 0.0010.

# Reporting Summary

## Statistics

For all statistical analyses, confirm that the following items are present in the figure legend, table legend, main text, or Methods section.

| n/a | Confirmed | |
|---|---|---|
| ☐ | ☒ | The exact sample size (*n*) for each experimental group/condition, given as a discrete number and unit of measurement |
| ☐ | ☒ | A statement on whether measurements were taken from distinct samples or whether the same sample was measured repeatedly |
| ☐ | ☒ | The statistical test(s) used AND whether they are one- or two-sided <br> *Only common tests should be described solely by name; describe more complex techniques in the Methods section.* |
| ☒ | ☐ | A description of all covariates tested |
| ☐ | ☒ | A description of any assumptions or corrections, such as tests of normality and adjustment for multiple comparisons |
| ☐ | ☒ | A full description of the statistical parameters including central tendency (e.g. means) or other basic estimates (e.g. regression coefficient) AND variation (e.g. standard deviation) or associated estimates of uncertainty (e.g. confidence intervals) |
| ☐ | ☒ | For null hypothesis testing, the test statistic (e.g. *F*, *t*, *r*) with confidence intervals, effect sizes, degrees of freedom and *P* value noted <br> *Give P values as exact values whenever suitable.* |
| ☒ | ☐ | For Bayesian analysis, information on the choice of priors and Markov chain Monte Carlo settings |
| ☒ | ☐ | For hierarchical and complex designs, identification of the appropriate level for tests and full reporting of outcomes |
| ☒ | ☐ | Estimates of effect sizes (e.g. Cohen's *d*, Pearson's *r*), indicating how they were calculated |

*Our web collection on statistics for biologists contains articles on many of the points above.*

## Software and code

Policy information about availability of computer code

| Data collection | Biopac AcqKnowledge (v4.2, 4.5, and 5.0), flexiWare v8.2, Olympus FluoView software vFV31S-SW, Leica LAS AF v2.3.6, Prizmatix Pulser v2.3.1, lox2 software SCIREQ v2.10.5.28, Zen Blue software v2.6 and 3.2, ImSpector v7.1.4 |
|---|---|
| Data analysis | Biopac AcqKnowledge (v4.2, 4.5, and 5.0), flexiWare v8.2, lox2 software SCIREQ v2.10.5.28, Fiji ImageJ v1.52p, R (v4.1.3) using Seurat (v4.1.1), Cell Ranger (pipeline v3.1.0) Enrichr (https://maayanlab.cloud/Enrichr/ ; October 1, 2022), Microsoft Excel (Office 365), Prism v9 (GraphPad), BORIS v8.20.4 |

For manuscripts utilizing custom algorithms or software that are central to the research but not yet described in published literature, software must be made available to editors and reviewers. We strongly encourage code deposition in a community repository (e.g. GitHub). See the Nature Portfolio guidelines for submitting code & software for further information.

## Data

Policy information about availability of data

All manuscripts must include a data availability statement. This statement should provide the following information, where applicable:
- Accession codes, unique identifiers, or web links for publicly available datasets
- A description of any restrictions on data availability
- For clinical datasets or third party data, please ensure that the statement adheres to our policy

Source data used for figures are provided, and single-cell transcriptome data of NEBs (NCBI GEO) are publicly available (GEO accession number to be determined).

# Research involving human participants, their data, or biological material

Policy information about studies with [human participants or human data](). See also policy information about [sex, gender (identity/presentation), and sexual orientation]() and [race, ethnicity and racism]().

| | |
|---|---|
| Reporting on sex and gender | N/A |
| Reporting on race, ethnicity, or other socially relevant groupings | N/A |
| Population characteristics | N/A |
| Recruitment | N/A |
| Ethics oversight | N/A |

Note that full information on the approval of the study protocol must also be provided in the manuscript.

# Field-specific reporting

Please select the one below that is the best fit for your research. If you are not sure, read the appropriate sections before making your selection.

☒ Life sciences  ☐ Behavioural & social sciences  ☐ Ecological, evolutionary & environmental sciences

For a reference copy of the document with all sections, see [nature.com/documents/nr-reporting-summary-flat.pdf]()

# Life sciences study design

All studies must disclose on these points even when the disclosure is negative.

| | |
|---|---|
| Sample size | Sample sizes were determined based on previous expertise and publications in our field (for example, PMID: 26855425, 28360327, 28002412, 36750092). Exact sample sizes are described in each figure legend or Methods. |
| Data exclusions | For behavioral analysis in Extended Data Fig. 7, occasional animals (3/24) tested were excluded because the animal was oriented away from the camera for most of the session, hindering accurate behavioral scoring. Exclusions were determined by a genotype-blind investigator. |
| Replication | All replicates were biological, unless otherwise indicated. All figures depicting representative images were independently replicated at least twice, but typically three or more times, and details are described in figure legends. |
| Randomization | Animals were randomly assigned to experimental cohorts, based on genotyping and age-matching. |
| Blinding | Investigators were genotype-blind to group allocations for plethysmography, physiological experiments, and behavioral analysis associated with Figures 3, 4, 5, and Extended Data Fig. 7. Blinding for experiments involving nerve transections, optogenetics, and comparisons of airway stimuli within a cohort was not possible as the same investigator applied the perturbation and recorded the response, and the perturbations are necessarily apparent in recordings for analysis. |

# Reporting for specific materials, systems and methods

We require information from authors about some types of materials, experimental systems and methods used in many studies. Here, indicate whether each material, system or method listed is relevant to your study. If you are not sure if a list item applies to your research, read the appropriate section before selecting a response.

## Materials & experimental systems

| n/a | Involved in the study |
|---|---|
| ☐ | ☒ Antibodies |
| ☒ | ☐ Eukaryotic cell lines |
| ☒ | ☐ Palaeontology and archaeology |
| ☐ | ☒ Animals and other organisms |
| ☒ | ☐ Clinical data |
| ☒ | ☐ Dual use research of concern |
| ☒ | ☐ Plants |

## Methods

| n/a | Involved in the study |
|---|---|
| ☒ | ☐ ChIP-seq |
| ☐ | ☒ Flow cytometry |
| ☒ | ☐ MRI-based neuroimaging |

# Antibodies

| | |
|---|---|
| Antibodies used | Primary Antibodies: anti-NCAM1, 1:250, Cell Signaling Technology, 99746S; anti-GFP, 5 microgram/ml, Aves Labs, GFP-1020; anti-mCherry/RFP, 3 microgram/ml, OriGene Technologies, AB0040-200; anti-HB-EGF (human), 1:250, R&D Systems, AF-259-NA; anti-RFP, Rockland, 1:1000, Rockland, 600-401-379.<br><br>Secondary Antibodies: Jackson Immunoresearch: anti-Chicken IgG-Alexa fluor 488, anti-rabbit IgG-Cy3, anti-rabbit IgG Cy5, anti-goat IgG Cy5, anti-goat IgG Cy3; Secondary antibody catalog numbers are RRIDs AB_2340375, AB_2307443, AB_2340607, AB_2340415, AB_2307351, respectively |
| Validation | Primary and secondary antibodies are commercially available and validated by the manufacturers. In our previous work with anti-DTR, GFP, and RFP antibodies, background staining was not observed in wild type animals lacking antigen (PMID: 31747594, 32259485, 33278342, 36890237)<br><br>Manufacturer Validation and Quality Control Practices:<br><br>Cell Signaling Technologies: https://www.cellsignal.com/about-us/cst-antibody-validation-principles<br><br>Aves Labs: "Antibodies were analyzed by western blot analysis (1:5000 dilution) and immunohistochemistry (1:500 dilution) using transgenic mice expressing the GFP gene product. Western blots were performed using BlokHen® (Aves Labs) as the blocking reagent, and HRP-labeled goat anti-chicken antibodies (Aves Labs, Cat. #H-1004) as the detection reagent"<br>OriGene Technologies: "In 293HEK cells transfected with cds plasmid detects a band of 29 kDa by Western blot. This antibody (AB0040) recognizes very well tdTomato and does not recognize GFP (green fluorescent protein)"<br><br>R&D Systems: "Detects human HB-EGF in ELISAs and Western blots. In direct ELISAs, less than 1% cross reactivity with recombinant mouse HB-EGF is observed. In sandwich immunoassays, less than 0.1% cross-reactivity with recombinant human (rh) Amphiregulin, rhBetacellulin, rhEpiregulin, and recombinant mouse Epigen is observed. "<br><br>Rockland: "This product was prepared from monospecific antiserum by immunoaffinity chromatography using Red Fluorescent Protein (Discosoma) coupled to agarose beads followed by solid phase adsorption(s) to remove any unwanted reactivities. Expect reactivity against RFP and its variants: mCherry, tdTomato, mBanana, mOrange, mPlum, mOrange and mStrawberry. Assay by immunoelectrophoresis resulted in a single precipitin arc against anti-Rabbit Serum and purified and partially purified Red Fluorescent Protein (Discosoma).  No reaction was observed against Human, Mouse or Rat serum proteins."<br><br>Jackson Immunoresearch: Based on immunoelectrophoresis and/or ELISA, the antibody reacts with whole molecule of host Ig. It also reacts with the light chains of other host species immunoglobulins. No antibody was detected against non-immunoglobulin serum proteins. The antibody has been tested by ELISA and/or solid-phase adsorbed to ensure minimal cross-reaction with non-host species such as chicken, guinea pig, syrian hamster, goat, horse, human, mouse, rabbit and rat serum proteins, but it may cross-react with immunoglobulins from other species. Whole IgG antibodies are isolated as intact molecules from antisera by immunoaffinity chromatography. They have an Fc portion and two antigen binding Fab portions joined together by disulfide bonds and therefore they are divalent. The average molecular weight is reported to be about 160 kDa. The whole IgG form of antibodies is suitable for the majority of immunodetection procedures and is the most cost effective. |

# Animals and other research organisms

Policy information about studies involving animals; ARRIVE guidelines recommended for reporting animal research, and Sex and Gender in Research

| | |
|---|---|
| Laboratory animals | Animals were maintained under constant environmental conditions (23±1 degree C, 46±5% relative humidity) with food and water provided ad libitum in a 12-h light-dark cycle. All studies used adult male and female mice in comparable numbers from mixed genetic backgrounds and ages 6 to 24 weeks old. All CreER mice and control littermates received tamoxifen (Sigma T5648, 100 mg/kg, IP, sunflower oil, twice 48 hrs apart) at least 10 days prior to further experiments. Mice containing Cre- and Flp-dependent DTR alleles were a generous gift from Martyn Goulding, and Calca-egfp mice were purchased (GENSAT, RRID:MMRRC_011187-UCD). For Pvalb-t2a-Cre, only female Cre mice were used for husbandry due to reported germline recombination in male breeders (Jax, 012358); male and female offspring were used for experiments. Olfr78-p2a-Cre mice were generated by pronuclear injection of Cas9 protein, CRISPR sgRNAs targeting the Olfr78 locus 3' UTR, and a single strand DNA template containing a p2a-cre gene cassette with 150 bp homology arms into C57BL/6 embryos. Knock-in pups were screened by PCR analysis, and correct expression of the transgene was verified by RNA in situ hybridization. All Cre driver lines used are viable and fertile, and abnormal phenotypes were not detected. All other mice were purchased from Jackson Laboratory, or made in the lab and then deposited at Jackson Laboratory: Ascl1-CreERT2 (012882), Nkx2.1-ires-Flp (028577), Piezo2-egfp-ires-Cre (027719), inter-G-alpha-q-DREADD (26942), lsl-SALSA (31968), lsl-TdTomato (007914), snap25-Gcamp6s (25111), lsl-ChR2 (012569), C57BL/6J (000664), lsl-G-alpha-q-DREADD (026220), loxP-Piezo2 (027720), loxP-Piezo1 (029213), Vglut2-ires2-Flpo (030212), inter-Ai65 (021875), Vglut2-ires-Cre (016963), Npy1r-gfp-Cre (030544), P2ry1-ires-Cre (29284), Pvalb-t2a-Cre (012358), Crhr2-ires-Cre (33728), Npy2r-ires-Cre (29285), Calb1-ires2-Cre (28532), Phox2b-Cre (16223), Glp1r-ires-Cre (29283), Mc4r-2a-Cre (030759), Gpr65-ires-Cre (029282). |
| Wild animals | No wild animals were used |
| Reporting on sex | All studies used adult male and female mice in comparable numbers from mixed genetic backgrounds. |
| Field-collected samples | No field-collected samples were used. |

| Ethics oversight | All animal procedures followed ethical guidelines outlined in the NIH Guide for the Care and Use of Laboratory Animals, and all procedures were approved by the Institutional Animal Care and Use Committee at Harvard Medical School. |
|---|---|

Note that full information on the approval of the study protocol must also be provided in the manuscript.

# Flow Cytometry

## Plots

Confirm that:

☐ The axis labels state the marker and fluorochrome used (e.g. CD4-FITC).

☐ The axis scales are clearly visible. Include numbers along axes only for bottom left plot of group (a 'group' is an analysis of identical markers).

☐ All plots are contour plots with outliers or pseudocolor plots.

☐ A numerical value for number of cells or percentage (with statistics) is provided.

## Methodology

| Sample preparation | The sample preparation was performed based on previously described tissue dissociations for neuroendocrine and rare epithelial cells (PMID: 30069046, 30069044, 31585080, 36810133, 36469459). In brief, whole lungs below the trachea were harvested from 10 Calca-EGFP and 10 Ascl1-CreER; lsl-tdTomato mice (5-7 weeks old, equal male/female, 10 days after tamoxifen injection), pooled by strain, minced and incubated (60 min, 37°C) in oxygenated papain dissociation buffer (Worthington Biochemical LK003150). Residual tissue was mechanically dissociated through a 100 micron cell strainer, pelleted by centrifugation (400g, 7 min, 4C), washed, resuspended in red blood cell lysis buffer (150 mM NH4Cl, 10 mM NaCHO3, 0.1 mM EDTA) for 5 min, pelleted, and resuspended in FACS buffer (0.5% bovine serum albumin, 2 mM EDTA, PBS, 4C). Immune cells were depleted with anti-CD45 magnetic beads according to the manufacturer's instructions (Biolegend 480027), and the remaining cells were resuspended in viability buffer (TO-PRO-3 and CellTrace Violet, both 1:10,000, in RPMI 1640; Thermo Fisher, T3605, 65-0854-39, 11835030, respectively) prior to sorting by the Immunology Flow Cytometry Core facility (HMS). |
|---|---|
| Instrument | FACS Aria II (BD Bioscience) |
| Software | BD FACSDiva Software |
| Cell population abundance | Abundance of sorted cells were determined by hemocytometer after sorting for 10X Genomics scRNA-seq. Cell population abundance was subsequently determined by scRNA-seq analysis, as depicted in Fig. 5, Extended Data 9 and described in Methods. |
| Gating strategy | The gating strategy was performed as previously described in exemplary gating plots for isolating neuroendocrine and rare epithelial cells (PMID: 30069046, 30069044, 31585080, 36810133, 36469459). In brief, all events were gated by morphology and complexity (FSC-A vs. SSC-A) to avoid cellular debris, and singlets identified by scatter height-width discrimination (FSC-H vs FSC-W). For sorting gates, unstained, single-stain, and fluorescence minus one controls were used to determine positive and negative boundaries. Live cells were gated on exclusion of TOPRO3 (TOPRO3-negative) and inclusion of CellTrace Violet (CellTrace Violet-positive) prior to gating of fluorescence marker positive cells (RFP+ for Ascl1-CreER; Ai14 and GFP+ for Calca-Egfp, respectively) for sorting. |

☐ Tick this box to confirm that a figure exemplifying the gating strategy is provided in the Supplementary Information.

