## [Peer Review File · Nature]

Manuscript Title: A vagal reflex evoked by airway closure

Reviewer Comments & Author Rebuttals

Reviewer Reports on the Initial Version:

Referees' comments:

Referee #1 (Remarks to the Author):

Breathing involves sensing inspiration and expiration. Over or under inflation of the lungs are potentially serious and, accordingly, there are neural circuits in place that trigger protective responses. Here, Schappe and colleagues show that compression triggers gasping in mice. The authors use a combination of physiological recordings, imaging, single cell transcriptomics and genetic tools to identify the sensory cells and transduction molecule involved and demonstrate their necessity. Overall, the study is exciting and the data are convincing with clear effects. The findings largely support the model that the authors propose. The experiments are well designed and generally follow logically from each other, and the paper is easy to read.

Below I have some comments geared towards helping get this study into better shape for eventual publication in Nature. I have a few main experimental suggestions that would improve the paper from great to fantastic. I also detail below a few others points that if addressed would help clarify a few things.

Main comments:

1) The paper almost entirely depends on the authors' physiological definition of a "gasp" inferred from their recordings. It would be very helpful for them to show a video or some data that these events correspond to gasping and remove any doubts, no matter how small, that those events are biologically meaningful and/or not artifacts or methodologically linked to the stimulation method.

2) This paper would benefit from linking the imaging of vagal neuron activity with the other results. The genetic manipulations and physiology make specific predictions that could be tested with imaging. I don't expect the authors to do all the below experiments, but some effort to link the data derived from the two approaches is essential. It would strengthen the paper and convince me that their model and conclusions are correct.

2A) The authors could express GCaMP directly in Pvalb neurons, and show that Pvalb positive neurons exhibit the response properties inferred from manipulating them. For example, do Pvalb neurons show responses to compression but not stretch? This would show directly that the compression-sensitive neurons they imaged in the vglut2 cre gcamp mice are the cells responsible for closure induced gasping. They could also use this as an opportunity to look more carefully at other types of airway stimulation e.g. chemical stimuli, given they later raise the possibility the NEBs might respond to this stimulation.

2B) the authors should test if the vagal ganglion compression responses go away when the NEBs are

abated. I realize the genetics are complicated here. However the authors later say the Pvalb neurons themselves express piezo2 which means they could use the Piezo2-Cre allele in the NEB intersect mouse to drive the GCaMP expression in Pvalb neurons on the NEB abated background . Again, it would elegantly link the effects of NEB ablation on respiratory physiology with its effects on vagal activity.

2C) Equally, it would add quite a bit if they authors could show that neuron activity due to compression is lost after specific loss of Piezo in the NEBs but not neurons. Since Pvalb neurons express Piezo2, it is hard to believe all mechanosensitivity is lost in these neurons – the authors use the analogy with the Merkel cell complex yet in that case Piezo2 in the neurons is essential whereas Merkel expression is more modulatory. I recognize this is likely beyond what can reasonably be accomplished in a short time frame but perhaps the authors could comment on this.

3) As mentioned above, the authors claim their NEB-vagus interaction is analogous to Merkel cell - LTMR interaction. However, there is one really critical difference. Piezo2 loss in NEBs but not vagus appears to completely block compression induced gasping (and presumably vagal activity). Thus, NEB Piezo2 is enough by itself to maintain gasp responses. However, in the Merkel cell - LTMR interaction, Piezo2 can be deleted from LTMRs and LTMR activity is totally gone. So Merkel cells are not sufficient for LTMR activity. NEBs are apparently much more important for the activity of their neuronal partners than Merkel cells. The authors therefore undersell their data and create confusion by claiming their pathway is analogous to the Merkel cell- LTMR interaction, given these important differences.

4) Can the authors please comment on the following? If Piezo2 and Piezo1 are responsible for vagal responses to airway stretch, why don't the Pvalb neurons, which themselves express Piezo2, also respond to stretch? As mentioned about, what is Piezo2 function in these cells?

5) Authors use the NEB intersect mouse incorporating Piezo2-Cre for their experiments, before they describe discovering that Piezo2 is expressed in NEBs using RNAseq. I don't believe the authors are clairvoyant, so perhaps there's a better way to frame this in the writing?

6) Throughout the study, the sample sizes are small- a power analysis should be done and the “n” increased accordingly. For example, in Figure 1c. Figure 3b,f,g and Figure 4g,h there is enough variation in the control responses to suggest more data is needed. Hard to know how biologically important a change of 0-5 “gasps” in the controls to 0-0.5 “gasping”.

•

7) Fig1b While the airway suction recordings look to have returned to pre-stimulation state, the cuff and nebulizer stimulated animals don't see to return to baseline. Showing longer recordings demonstrating that the pressure fluctuations (rapid exhalation) can return to the pre-stimulation variations in pressure would be helpful

8) Please clarify the anesthesia methods used for each experiment. Do you see isoflurane induced gasping- something that is quite common in rodent long surgeries? Either too high, too fast or too long exposure to isoflurane will also induce gasping/ rhythmic gasping. This seems counter to the data here?

Minor:

-Check the figure legends correct correspond to the figures.

-Figure 1B Top and middle rows do not have pressure scales in the traces of physiological measurements.

-Fig1D- Please comment on the differences between severing different vagal branches, why does Glossopharyngeal affect hypoxia gasping while vagal trunk affects cuff induced gasping?-

-Fig2 – Please match the heights of the heat map in each functional group to that of fig2C proportions, or to make the height of each of the functional groups the exact same height. Please indicate the number of cells in each functional group represented.

-How were the 468 other cells that do not respond to any stimulation selected for inclusion? Do they just express Salsa? Is the proportion of non-responders meaningful?

- While ratiometric imaging is an innovative way to image vagal neurons, what does a negative $\Delta F/F$ mean?. Do the authors believe this is some sort of suppression of calcium or are negative an artifact of the analyses? If the later, perhaps make all negative values the same color as zero to avoid confusion?

- Please detail the proportions of each functional group per mouse. Plotted in a histogram to see if these proportions are uniform across all 7 animals (compression, multimodal, inflation only and other cells)

-Extended Fig 2- correct the stretching of each functional group the heatmaps.

-Fig3f- Ns vary. In the group that have PV:DTR ablated vagal neurons, only 2 animals gasped less than the controls? The controls here seem to gasp less (~2-3 gasps per min) than they did in Figure 1 (~5gasps per min). Why are there differences? Does chemical nebulized gasping change in these PV:DTA animals as well?

Referee #2 (Remarks to the Author):

Congratulations to the authors for this novel and comprehensive study. This manuscript highlights several important discoveries for breathing physiology, such as a role for the NEBs in the modulation of breathing, the identification of stretch vs. compression specific vagal sensory pathways, and a cellular / molecular mechanism for the breathing response to atelectasis. These results modernize a decades old lung-deflation reflex and will establish a foundation for future studies, like the dissection of the brainstem mechanisms for the breathing response and how these sensory signals may convey sensations like breathlessness to higher brain centers. These data have the potential to inform multiple medical fields that depend on careful control of lung ventilation, such as critical care / lung ventilation. I believe this manuscript will be of broad interest to the readers of Nature from neuroscientists, physiologists, cellular biologists, and clinicians.

A general comment is that the breath type that is described here as a “gasp” is equivalent to a “sigh” breath. One of the important physiological functions of a sigh is to reverse alveolar collapse which is the proposed purpose of the “gasp” in this manuscript. Additionally, the authors propose that the same sigh generating brainstem mechanisms are likely used to generate the “gasps”. Others have used the term gasp during the autoresuscitation response (just one example, Dosumu-Johnson 2018) which is not investigated here. I believe that the importance and novelty of the story does not depend upon this apparent sigh breath now being redescribed as a gasp. In fact, this manuscript will significantly strengthen and an ongoing literature on the physiological importance of and cellular and molecular mechanism of a sigh breath (Li, 2016; Li, 2020; Yao, 2023; etc.). Additionally, the manuscript is focused on a single breath type, but many of the data point to a larger respiratory response to lung restriction that is being directed by the NEBs (such as a general increase in breathing effort), and in this case, a “gasp” is just one of the components.

I have several comments to consider that are mostly centered around 1) a more comprehensive representation of the data, 2) a more thorough description of the breathing response, and 3) a direct demonstration of the NEB to PARVB sensory signaling pathway.

1. As presented throughout the manuscript, the data is portrayed as a single example trace and then an average for each animal. However, what is missing is the variability in the response and the kinetics of the time-to-gasp after the initiation of each trial (even the number of trials to calculate the average for each animal is unclear from the methods). For example, do the gasps occur at the offset of each compression trial (as appears in Fig 1b, 4g, 5e, etc.). This data should be included and can easily be done in many ways, such as raster plots of when the gasp occurs in many trials. Below are examples where this should be done.

- a. Figure 1b – kinetics and variability of the gasp response in each experimental paradigm.
- b. Figure 2d – R_c/R_i ratios for each cell over many trials
- c. Figure 3b – trials of optogenetic stimulation
- d. Extended Data 6 (and other pleth data) – the distribution of these parameter for each breath during the recording as opposed to just a single average value over 30 min.

2. The authors should include a more comprehensive characterization of the breathing response to lung restriction. Most importantly, the authors should directly measure the activity of a breathing

muscle, like the diaphragm, instead of inferring the changes in breathing by esophageal pressure. One should not assume that the only fluctuation in esophageal pressure is due to breathing. Also, a direct measure of the diaphragm could enable a more comprehensive characterization of the breathing response. For example, does the strength of contraction increase in time over the course of the restriction (this appears to be the case in many of the example traces). A more complete characterization may provide an explanation for the differences between example traces, such as Figure 1b and 4l. If a breathing response beyond a gasp is defined, this would reveal an important role for the NEBs in generating an entire response to lung restriction or to generating larger breaths in general to lung deflation which is supported by 1) the excitation of NEBs increasing PIF (4e) and 2) the loss of function which leads to a decrease in lung compliance during mechanical ventilation (Extended Data 7).

3. Connectivity between the PVALB sensory neurons and the NEBs is inferred based on the similar responses to breathing upon modulation of either cell type, but this connectivity should be demonstrated. For example, it is not shown if PARVB neurons sense lung compression and if so, do they also respond to inflation? Does the complete vagal lung compression signal go away upon ablation of PVALB neurons? Are PVALB neurons activated upon chemogenetic excitation of NEBs? Are PARVB neurons required for the HBR?

4. The lung compression assays in each experiment can be more carefully assayed by controlling atelectasis with variations in mechanical ventilation (like changing PEEP). When done this way, it would provide more direct evidence for lung restriction (as opposed to chest restriction) as the signal to induce a gasp and would enable dose response curves to be conducted. For example, in Figure 1, do changes in PEEP correlate with the frequency or probability of a gasp or the changes in breathing measured by activity of a breathing muscle. Also, do these changes in PEEP correlate with the extent of vagal nerve activation in Figure 2?

5. Gasps are reported as per minute when the assays are over the course of 10 seconds. This is a large extrapolation.

6. The changes in compliance in Extended Data 1 could be explained by restriction of the chest wall and so are an indirect measure of lung volume. This should be acknowledged.

7. Urethane is reported as 1.6 mg/kg. Is this correct? A standard dose is 1.6 mg/g.

Referee #3 (Remarks to the Author):

Our lungs respond to sudden airway obstruction, airway lumen closure, and hypoxia by increasing the effort of breathing, including gasping. In a recent study, Schappe et al. addressed a fundamental question about how the lung mechanically senses and responds to sudden airway closure involved in gasping in mice. They identified that neuroendocrine cell clusters (neuroepithelial bodies, NEBs) sense airway closure and send signals back to the central nervous system via PVALB+ vagal neurons.

Specifically, this manuscript advanced our understanding of lung physiology in two aspects: (1) identifying the PVALB+ vagal neurons specifically involved in the sensory aspect of airway closure and gasping; and (2) revealing the physiological function of PIEZO2-expressing neuroendocrine cells. Pulmonary neuroendocrine cells have long been proposed to be involved in mechanical sensation in the lungs. Recent gene expression profiling of neuroendocrine cells in human and mouse lungs revealed that they express force-sensing channel PIEZO2. Schappe et al. directly addressed this hypothesis via thorough investigations in mice. Physiological studies in multiple genetically modified mouse strains are particularly impressive. The data strongly support the conclusions in the manuscript.

Specific comments:

1. It is easy to understand that stretches on NEBs open the PIEZO2 channel. What is the case of squeeze on NEBs? What type of neurotransmitters and neuropeptides are involved in the NEBs-Pvalb+ vagal neuron interaction? These are all unresolved questions in this paper. The authors might wish to expand the discussion related to these questions.

2. Description of Figure 2 in the main text (First paragraph, page 7) and figure legend. The description of the fraction of different types of vagal neurons in the main text does not match well with Figure 2, which includes several key elements. The authors should consider linking the data in the text more clearly to Figure 2.

3. The authors might also want to describe how the cells (each row) in Figure 2b and Extended Data Figure 2a are grouped (manually or clustering using R) in the figure legend.

4. Based on the first paragraph of Page 8, Pvalb+ vagal neurons are a subtype of P2ry1+ vagal neurons. In authors' previous study, P2ry1+ vagal neurons including J3, NP16, NP17, NP19 and NP26 clusters (Figure 2B of Prescott et al., 2020), two of which express Pvalb (NP17 & NP19) based on Figure 3c. It would be great if the authors could point out the specific Pvalb+ vagal type they are referring to. I speculate that NP17 and NP19 Pvalb+ neurons might have different functions.

5. In the discussion, the authors should recognize that these findings in mice have uncertain applicability to human and other large mammals both in terms of the distribution of NEBs and frequency of solitary neuroendocrine cells. It would also be very interesting to know whether PVALB+ (or CRHR2+) vagal neurons innervate human distal (bronchioles or alveolar) NEBs.

6. Figure 3i, miss the label for the bottom right image.

7. In the last 2 lines of Page 11, the authors wrote that “Solitary neuroendocrine cells were only sparsely labeled in Piezo2-ires-Cre; Nkx2.1-Flpo mice”. I cannot find such data. The extended data Figure 5a has an image of larynx from the Nkx2.1-Piezo2Cre mice, but I cannot see any fluorescent dots.

8. The finding of PIEZO2 in NEBs (related to the first paragraph of Page 14 and Figure 5c). Recent gene expression profiling (Ouadah et al., 2019, Cell; Kuo et al. eLife 2022; LungMAP) of human and mouse airways showed that neuroendocrine cells express PIEZO2 and other sensory receptors. In addition, in vitro physiological studies [Pan et al., 2006 (PMID: 16100287); Robrecht Lembrechts et al., 2012 (PMID: 22461428)] suggest that PNECs sense mechanical force. I would not directly claim that “underlying sensory receptors are unclear”. The knowledge gap should be the lack of physiological evidence in vivo.

9. The authors might move the heatmap in Figure 5b to the extended data. It is very interesting to show subtypes of neuroendocrine cells and their markers in a heatmap. The current heatmap does not include any gene names.

10. Figure 5d. I don't see quantification graphs (as figure legend claimed) for Piezo2 in situ hybridization.

11. In the extended data figure 8, I would not consider Epcam and Nkx2.1 as markers for the neuroendocrine cells since they are expressed in other epithelial cell types of the lungs based on other studies and single-cell RNA sequencing data.

Author Rebuttals to Initial Comments:

Referees' comments:

Referee #1 (Remarks to the Author):

Breathing involves sensing inspiration and expiration. Over or under inflation of the lungs are potentially serious and, accordingly, there are neural circuits in place that trigger protective responses. Here, Schappe and colleagues show that compression triggers gasping in mice. The authors use a combination of physiological recordings, imaging, single cell transcriptomics and genetic tools to identify the sensory cells and transduction molecule involved and demonstrate their necessity. Overall, the study is exciting and the data are convincing with clear effects. The findings largely support the model that the authors propose. The experiments are well designed and generally follow logically from each other, and the paper is easy to read.

Below I have some comments geared towards helping get this study into better shape for eventual publication in Nature. I have a few main experimental suggestions that would improve the paper from great to fantastic. I also detail below a few others points that if addressed would help clarify a few things.

Thank you for your helpful and supportive comments. We address each of your questions, as detailed below.

Main comments:

1) The paper almost entirely depends on the authors' physiological definition of a "gasp" inferred from their recordings. It would be very helpful for them to show a video or some data that these events correspond to gasping and remove any doubts, no matter how small, that those events are biologically meaningful and/or not artifacts or methodologically linked to the stimulation method.

We now include a new video that illustrates a typical gasping response (Supplementary Video 2), with behavioral recordings performed in a plethysmography chamber to provide simultaneous analysis of respiratory changes. While analyzing behavioral videos, we additionally observed that chemogenetic stimulation of NEBs in freely behaving mice caused respiratory distress with a characteristic hunching posture, labored breathing, and immobility, as might be expected for a mouse experiencing air hunger. In addition to providing relevant videos (Supplementary Video 2), we also quantified behavioral observations and report them in Extended Data Fig. 7b-c. Finally, we added additional physiological evidence of gasping, as EMG recordings of intercostal muscles and diaphragm show enhanced activity of respiratory muscles during gasping events.

2) This paper would benefit from linking the imaging of vagal neuron activity with the other results. The genetic manipulations and physiology make specific predictions that could be tested with imaging. I don't expect the authors to do all the below experiments, but some effort to link the data derived from the two approaches is essential. It would strengthen the paper and convince me that their model and conclusions are correct.

Thank you for these suggestions, we have done several additional experiments to address each of these points below.

2A) The authors could express GCaMP directly in Pvalb neurons, and show that Pvalb positive neurons exhibit the response properties inferred from manipulating them. For example, do Pvalb neurons show responses to compression but not stretch? This would show directly that the compression-sensitive neurons they imaged in the *vglut2 cre gcamp* mice are the cells responsible for closure induced gasping. They could also use this as an opportunity to look more carefully at other types of airway stimulation e.g. chemical stimuli, given they later raise the possibility the NEBs might respond to this stimulation.

We have now performed calcium imaging in mice with genetic labels expressed in Pvalb neurons. In addition, we added airway suction as a stimulus in our imaging paradigm, and observed that airway suction activates fewer vagal neurons than airway compression. This may be because airway compression more effectively stimulates NEBs and/or compression also stimulates other thoracic mechanoreceptors in addition to the airway closure receptors. Importantly, 1) most neurons responsive to airway suction also responded to airway compression, and 2) most neurons responsive to both airway suction and compression expressed PVALB (13/14; 92.9%).

2B) the authors should test if the vagal ganglion compression responses go away when the NEBs are abated. I realize the genetics are complicated here. However the authors later say the Pvalb neurons themselves express *piezo2* which means they could use the *Piezo2-Cre* allele in the NEB intersect mouse to drive the GCaMP expression in Pvalb neurons on the NEB abated background. Again, it would elegantly link the effects of NEB ablation on respiratory physiology with its effects on vagal activity.

We performed new whole nerve electrophysiology experiments to measure vagal afferent responses to airway suction. We observed robust nerve responses to airway closure in wild type mice, but no responses in NEB-ABLATE mice. These observations are now described in Figure 5, and provide a direct link between NEB function and vagal nerve responses to airway closure, strongly supporting our mechanistic model.

2C) Equally, it would add quite a bit if they authors could show that neuron activity due to compression is lost after specific loss of Piezo in the NEBs but not neurons. Since Pvalb neurons express Piezo2, it is hard to believe all mechanosensitivity is lost in these neurons – the authors use the analogy with the Merkel cell complex yet in that case Piezo2 in the neurons is essential whereas Merkel expression is more modulatory. I recognize this is likely beyond what can reasonably be accomplished in a short time frame but perhaps the authors could comment on this.

We also performed whole vagus nerve electrophysiology to measure airway closure responses in mice with genetic deletion of PIEZO2 in NEBs. As with NEB ablation, we observe that knockout of PIEZO2 in NEBs eliminates vagal responses to airway closure, while responses to airway stretch remained intact. These new findings are now reported in Figure 5. We note that PIEZO2 is expressed in about 1/3rd of vagal Pvalb neurons based on transcriptomic data, and its expression directly in Pvalb neurons is not required for gasping to airway closure. We agree that there are functional differences between NEBs and Merkel cells and the analogy is only partial, so we have revised the text accordingly.

3) As mentioned above, the authors claim their NEB-vagus interaction is analogous to Merkel cell - LTMR interaction. However, there is one really critical difference. Piezo2 loss in NEBs but not vagus appears to completely block compression induced gasping (and presumably vagal activity). Thus, NEB Piezo2 is enough by itself to maintain gasp responses. However, in the Merkel cell - LTMR interaction, Piezo2 can be deleted from LTMRs and LTMR activity is totally gone. So Merkel cells are not sufficient for LTMR activity. NEBs are apparently much more important for the activity of their neuronal partners than Merkel cells. The authors therefore undersell their data and create confusion by claiming their pathway is analogous to the Merkel cell- LTMR interaction, given these important differences.

We agree completely, and have updated the text to highlight differences and similarities between NEBs and Merkel cells. In the discussion, we now say: "We suggest that NEBs are somewhat similar to Merkel cells in the skin involved in touch sensation in that they are PIEZO2-utilizing epithelial cells that communicate with peripheral sensory neurons. However, there are residual touch responses following PIEZO2 knockout in Merkel cells, while airway closure responses reported here are absent following PIEZO2 knockout in NEBs." We also toned down the analogy in the abstract and say "Like Merkel cells involved in touch sensation, NEBs are PIEZO2-expressing epithelial cells, and moreover, are critical for an aspect of lung mechanosensation."

4) Can the authors please comment on the following? If Piezo2 and Piezo1 are responsible for

vagal responses to airway stretch, why don't the Pvalb neurons, which themselves express Piezo2, also respond to stretch? As mentioned about, what is Piezo2 function in these cells?

Terminals responsive to airway stretch and airway closure are structurally quite distinct. While NEBs and Pvalb neurons mediate closure responses, we strengthened data to show that they are dispensable for airway stretch responses (Extended Data Fig. 4g). Instead, airway stretch responses are thought to involve terminals in smooth muscle. These data suggest that the terminal structure and anatomical localization of Piezo2 is essential towards determining what a neuron detects. There is precedence for cells that express Piezo2, but do not require it for sensation, such as hair cells in the auditory system. Moreover, Piezo2 is only expressed in a subset of PVALB neurons.

5) Authors use the NEB intersect mouse incorporating Piezo2-Cre for their experiments, before they describe discovering that Piezo2 is expressed in NEBs using RNAseq. I don't believe the authors are clairvoyant, so perhaps there's a better way to frame this in the writing?

We prefer to introduce the single-cell seq analysis after introducing the genetic tools, but recognize that this was not the order of experimentation. We hope it is ok to simply refer ahead in the manuscript with the phrase "see single cell analysis of NEBs below", but can re-work the text if the reviewer feels strongly.

6) Throughout the study, the sample sizes are small- a power analysis should be done and the "n" increased accordingly. For example, in Figure 1c. Figure 3b,f,g and Figure 4g,h there is enough variation in the control responses to suggest more data is needed. Hard to know how biologically important a change of 0-5 "gasps" in the controls to 0-0.5 "gasping".

We have increased the sample size in all figures mentioned.

7) Fig1b While the airway suction recordings look to have returned to pre-stimulation state, the cuff and nebulizer stimulated animals don't seem to return to baseline. Showing longer recordings demonstrating that the pressure fluctuations (rapid exhalation) can return to the pre-stimulation variations in pressure would be helpful

We added a panel in Extended Data Fig. 1g to show full recovery and return to baseline, which typically occurs in 10 seconds or less after airway compression.

8) Please clarify the anesthesia methods used for each experiment. Do you see isoflurane induced gasping- something that is quite common in rodent long surgeries? Either too high, too fast or too long exposure to isoflurane will also induce gasping/ rhythmic gasping. This seems counter to the data here?

We clarified the anesthetics used in the methods. Initial experiments in Extended Data Figure 1 led us to adopt urethane as the anesthetic of choice, and we did not use isoflurane for any subsequent experiments in the manuscript. Gasps were defined quantitatively by comparisons

to baseline, and mice under isoflurane had high baseline tidal volumes that perhaps obscured further increases.

Minor:

-Check the figure legends correct correspond to the figures.

Thanks- we doublechecked this. Please let us know if we missed something though!

-Figure 1B Top and middle rows do not have pressure scales in the traces of physiological measurements.

We added these.

-Fig1D- Please comment on the differences between severing different vagal branches, why does Glossopharyngeal affect hypoxia gasping while vagal trunk affects cuff induced gasping?-

Hypoxia-induced gasping is mediated by different sensory neurons that innervate the carotid body, and the carotid sinus nerve is part of the glossopharyngeal nerve. We think these findings provide a nice control that the motor arm of the gasping reflex is intact after genetic and surgical manipulations.

-Fig2 – Please match the heights of the heat map in each functional group to that of fig2C proportions, or to make the height of each of the functional groups the exact same height. Please indicate the number of cells in each functional group represented.

We edited the heat map in all figures as requested. We previously depicted all neurons, which as the reviewer noted resulted in a shrinking of data height for unresponsive neurons, since most neurons fell in this class. We have now kept data height similar for each neurons, and to do so, depicted only some randomly selected non-responsive neurons. The total numbers imaged are in the figures and legends.

-How were the 468 other cells that do not respond to any stimulation selected for inclusion? Do they just express Salsa? Is the proportion of non-responders meaningful?

Prior experiments showed that vagal nerve stimulation was an effective method for ensuring neuron viability/responsiveness. In previous publications involving this preparation from our lab (i.e. Williams 2016, Prescott 2020), the majority (~66%) of indicator-expressing neurons are responsive to vagal nerve stimulation, and moreover unresponsive neurons could be readily identified based on strong and unvarying GCaMP fluorescence relative to neighboring cells at baseline. In these experiments, we included all Salsa-positive neurons that lacked such strong and unvarying fluorescence. We updated the Methods to describe this more clearly. We do think the proportion of non-responders is meaningful to understand the abundance of airway closure neurons in the larger context of vagal functionality.

- While ratiometric imaging is an innovative way to image vagal neurons, what does a negative $\Delta F/F$ mean?. Do the authors believe this is some sort of suppression of calcium or are negative values an artifact of the analyses? If the later, perhaps make all negative values the same color as zero to avoid confusion?

We prefer keeping a report on negative values (as measured relative to baseline) as we asked whether airway stretch neurons might decrease activity during airway closure. We instead observed that the dominant response was increased activity in a discrete neuronal cohort, so there is not much emphasis of the negative scale, but we thought it was instructive to include.

- Please detail the proportions of each functional group per mouse. Plotted in a histogram to see if these proportions are uniform across all 7 animals (compression, multimodal, inflation only and other cells)

We added these data as a panel in Extended Data Figure 2a, which confirm that response distributions are rather conserved across mice. We note that average response percentages in Figure 2 are derived from all neurons imaged.

-Extended Fig 2- correct the stretching of each functional group the heatmaps.

We did this, as described for Figure 2 above.

-Fig3f- Ns vary. In the group that have PV:DTR ablated vagal neurons, only 2 animals gasped less than the controls? The controls here seem to gasp less (~2-3 gasps per min) than they did in Figure 1 (~5 gasps per min). Why are there differences? Does chemical nebulized gasping change in these PV:DTA animals as well?

We increased sample size and new data matches all of our prior conclusions. We also added new experiments involving nebulized methacholine in Extended Data 4e.

Referee #2 (Remarks to the Author):

Congratulations to the authors for this novel and comprehensive study. This manuscript highlights several important discoveries for breathing physiology, such as a role for the NEBs in the modulation of breathing, the identification of stretch vs. compression specific vagal sensory pathways, and a cellular / molecular mechanism for the breathing response to atelectasis. These results modernize a decades old lung-deflation reflex and will establish a foundation for future studies, like the dissection of the brainstem mechanisms for the breathing response and how these sensory signals may convey sensations like breathlessness to higher brain centers. These data have the potential to inform multiple medical fields that depend on careful control of lung ventilation, such as critical care / lung ventilation. I believe this manuscript will be of broad interest to the readers of Nature from neuroscientists, physiologists, cellular biologists, and clinicians.

A general comment is that the breath type that is described here as a “gasp” is equivalent to a “sigh” breath. One of the important physiological functions of a sigh is to reverse alveolar collapse which is the proposed purpose of the “gasp” in this manuscript. Additionally, the authors propose that the same sigh generating brainstem mechanisms are likely used to generate the “gasps”. Others have used the term gasp during the autoresuscitation response (just one example, Dosumu-Johnson 2018) which is not investigated here. I believe that the importance and novelty of the story does not depend upon this apparent sigh breath now being redescribed as a gasp. In fact, this manuscript will significantly strengthen and an ongoing literature on the physiological importance of and cellular and molecular mechanism of a sigh breath (Li, 2016; Li, 2020; Yao, 2023; etc.). Additionally, the manuscript is focused on a single breath type, but many of the data point to a larger respiratory response to lung restriction that is being directed by the NEBs (such as a general increase in breathing effort), and in this case, a “gasp” is just one of the components.

Thank you for your helpful response and thoughtful discussion. We fully agree with the reviewer that a gasp is just one component of an airway closure-evoked motor program, and we provide new data about behavioral responses, respiratory muscle activity, and changes in lung mechanics. In accordance with the reviewer's sentiment, we have removed the word gasp from the title consistent with the broader physiological context. In the main text, we concluded that these respiratory events were gasps using a collection of physiological approaches, and this descriptor is intuitive to the stimulus applied. We added discussion in the revised manuscript and to connect these findings to the strong literature referenced on sighs. Thank you for this helpful discussion.

I have several comments to consider that are mostly centered around 1) a more comprehensive representation of the data, 2) a more thorough description of the breathing response, and 3) a direct demonstration of the NEB to PARVB sensory signaling pathway.

1. As presented throughout the manuscript, the data is portrayed as a single example trace and then an average for each animal. However, what is missing is the variability in the response and the kinetics of the time-to-gasp after the initiation of each trial (even the number of trials to calculate the average for each animal is unclear from the methods). For example, do the gasps occur at the offset of each compression trial (as appears in Fig 1b, 4g, 5e, etc.). This data should be included and can easily be done in many ways, such as raster plots of when the gasp occurs in many trials. Below are examples where this should be done.

- a. Figure 1b – kinetics and variability of the gasp response in each experimental paradigm.
- b. Figure 2d – R_c/R_i ratios for each cell over many trials
- c. Figure 3b – trials of optogenetic stimulation
- d. Extended Data 6 (and other pleth data) – the distribution of these parameter for each breath during the recording as opposed to just a single average value over 30 min.

a. We have now added raster plots (Extended Data Fig. 1d) indicating gasping events over time for each stimulus in Figure 1b.

b. We now include data (Extended Data Fig. 2b) indicating the activity of each neuron from a representative mouse over repeated trials. We think that providing the complete activity trace is more informative than a single Rc/Ri value. We also included a new figure panel in Extended Data Fig. 2a indicating minimal animal-to-animal variability in the relative distribution of responses.

c. We have now added raster plots (Extended Data Fig. 3c) indicating gasping events over time for optogenetic stimulation in Figure 3b.

d. In Extended Data 6c, we added a collection of histograms depicting breathing parameters on a per breath basis for individual animals before/after chemogenetic stimulation of NEBs. Our plethysmography experiments involved an extended acclimation period which helped minimize variability due to behavioral adaptation over the 30 minute trial, and these data strongly support our findings reporting average values over 30 min.

2. The authors should include a more comprehensive characterization of the breathing response to lung restriction. Most importantly, the authors should directly measure the activity of a breathing muscle, like the diaphragm, instead of inferring the changes in breathing by esophageal pressure. One should not assume that the only fluctuation in esophageal pressure is due to breathing. Also, a direct measure of the diaphragm could enable a more comprehensive characterization of the breathing response. For example, does the strength of contraction increase in time over the course of the restriction (this appears to be the case in many of the example traces). A more complete characterization may provide an explanation for the differences between example traces, such as Figure 1b and 4I. If a breathing response beyond a gasp is defined, this would reveal an important role for the NEBs in generating an entire response to lung restriction or to generating larger breaths in general to lung deflation which is supported by 1) the excitation of NEBs increasing PIF (4e) and 2) the loss of function which leads to a decrease in lung compliance during mechanical ventilation (Extended Data 7).

Thank you for this point. We have now performed EMG recordings of intercostal muscles and the diaphragm during airway closure. This was effectively done for airway suction and methacholine administration, but such recordings were not feasible during thoracic compression due to the presence of the cuff. These findings are presented in Extended Data Figure 1, and show that all gasp events are similarly detected by measurements of esophageal pressure, intercostal muscle EMG and diaphragm EMG. We also agree with the reviewer that gasps are part of more comprehensive motor program, with recruitment of respiratory muscles helping to enhanced airflow. We think these new findings support and strengthen our original claims.

3. Connectivity between the PVALB sensory neurons and the NEBs is inferred based on the similar responses to breathing upon modulation of either cell type, but this connectivity should be demonstrated. For example, it is not shown if PARVB neurons sense lung compression and if so, do they also respond to inflation? Does the complete vagal lung compression signal go away upon ablation of PVALB neurons? Are PVALB neurons activated upon chemogenetic excitation of NEBs? Are PARVB neurons required for the HBR?

We performed a large number of additional experiments to demonstrate the direct connectivity between airway closure, NEBs, and PVALB neurons.

First, we performed calcium imaging in mice with genetic labels expressed in Pvalb neurons. In addition, we added airway suction as a stimulus in our imaging paradigm, and observed that airway suction activates fewer vagal neurons than airway compression. This may be because airway compression more effectively stimulates NEBs and/or compression also stimulates other thoracic mechanoreceptors in addition to the airway closure receptors. Importantly, 1) most neurons responsive to airway suction also responded to airway compression, and 2) most neurons responsive to both airway suction and compression expressed PVALB (13/14; 92.9%).

Second, we performed new whole nerve electrophysiology experiments to measure vagal afferent responses to airway suction. We observed robust nerve responses to airway closure in wild type mice, but no responses in NEB-ABLATE mice. These observations are now described in Figure 5, and provide a direct link between NEB function and vagal nerve responses to airway closure, strongly supporting our mechanistic model. We note that chemogenetic activation of NEBs is not technically possible while imaging PVALB neurons, as too many Cre alleles would be needed in parallel. For this reason, we pursued whole nerve electrophysiology experiments in the context of targeted NEB ablation. Together, we now show the close anatomical relationship of NEBs and PVALB neurons, genetic ablation data showing a suite of neuronal and physiological responses to airway closure that require both NEBs and PVALB neurons, and calcium imaging data that PVALB neurons are acutely activated by closure.

Third, we also performed whole vagus nerve electrophysiology to measure airway closure responses in mice with genetic deletion of PIEZO2 in NEBs. As with NEB ablation, we observe that knockout of Piezo2 in NEBs eliminates vagal responses to airway closure, while responses to airway stretch remained intact. These new findings are now reported in Figure 5.

Fourth, in new data, we observed that the Hering-Breuer reflex is fully intact following ablation of PVALB neurons (Extended Data Fig. 4g), in addition to NEBs as shown in the original submission (Fig. 4j).

4. The lung compression assays in each experiment can be more carefully assayed by controlling atelectasis with variations in mechanical ventilation (like changing PEEP). When done this way, it would provide more direct evidence for lung restriction (as opposed to chest restriction) as the signal to induce a gasp and would enable dose response curves to be conducted. For example, in Figure 1, do changes in PEEP correlate with the frequency or probability of a gasp or the changes in breathing measured by activity of a breathing muscle. Also, do these changes in PEEP correlate with the extent of vagal nerve activation in Figure 2?

Precise control of PEEP requires mechanical ventilation with paralytics or an open-chest preparation where airway closure-induced gasps are not observed due to elimination of the contribution of respiratory muscles to breathing. We performed experiments involving coarser PEEP manipulations by ventilating spontaneously breathing animals, which we note do not

reflect the natural progression of small airway closure during atelectasis. We did observe increased gasping after lowering end-expiratory pressure, as the reviewer predicts, but hesitate to describe this experiment in the paper since such PEEP manipulations will not only cause atelectasis, but may cause secondary vagal activation through tissue damage, and secondary hypoxia which will promote gasping through NEB-independent mechanisms, as shown by Yao, 2023.

5. Gasps are reported as per minute when the assays are over the course of 10 seconds. This is a large extrapolation.

We prefer to leave this measure but can change the Y-axis scale if the reviewer feels strongly. We believe that gasps per minute is intuitive; prior studies have reported sighs per hour, and other breathing extrapolations like breaths per minute are pervasive. We also added raster plots for evoked gasps across animals and trials, which help highlight the frequency of these events.

6. The changes in compliance in Extended Data 1 could be explained by restriction of the chest wall and so are an indirect measure of lung volume. This should be acknowledged.

We agree with the reviewer, and add clarification of this point in the text.

7. Urethane is reported as 1.6 mg/kg. Is this correct? A standard dose is 1.6 mg/g.

Thank you for catching this typo.

Referee #3 (Remarks to the Author):

Our lungs respond to sudden airway obstruction, airway lumen closure, and hypoxia by increasing the effort of breathing, including gasping. In a recent study, Schappe et al. addressed a fundamental question about how the lung mechanically senses and responds to sudden airway closure involved in gasping in mice. They identified that neuroendocrine cell clusters (neuroepithelial bodies, NEBs) sense airway closure and send signals back to the central nervous system via PVALB+ vagal neurons.

Specifically, this manuscript advanced our understanding of lung physiology in two aspects: (1) identifying the PVALB+ vagal neurons specifically involved in the sensory aspect of airway closure and gasping; and (2) revealing the physiological function of PIEZO2-expressing neuroendocrine cells. Pulmonary neuroendocrine cells have long been proposed to be involved in mechanical sensation in the lungs. Recent gene expression profiling of neuroendocrine cells in human and mouse lungs revealed that they express force-sensing channel PIEZO2. Schappe et al. directly addressed this hypothesis via thorough investigations in mice. Physiological studies in multiple genetically modified mouse strains are particularly impressive. The data strongly support the conclusions in the manuscript.

Thank you for your kind and helpful comments. We addressed each of your comments as detailed below.

Specific comments:

1. It is easy to understand that stretches on NEBs open the PIEZO2 channel. What is the case of squeeze on NEBs? What type of neurotransmitters and neuropeptides are involved in the NEBs-Pvalb+ vagal neuron interaction? These are all unresolved questions in this paper. The authors might wish to expand the discussion related to these questions.

Thank you for this point. We have expanded discussion of these points. One model is that airway closure prevents dissipation of inspiratory pressure across the entire pulmonary tree, leading to local pressure increases in conducting airways, where NEBs are enriched, and decreased counterpressure in alveoli. In the context of this model, Piezo2 would still act as stretch-activated channel rather than a squeeze-activated channel. It is enticing that a model involving mechanosensation at airway bottlenecks would also explain why some classically defined deflation receptors are also activated by large lung inflations, with transient activity (apparent rapid adaptation) potentially explained by a short-lived pressure increase in large conducting airways that soon disperses throughout the lungs. We have not explored neurotransmitters involved in NEB-PVALB neuron communication, as this paper instead focused on sensory mechanisms and physiological outputs. This could be the subject of future studies.

2. Description of Figure 2 in the main text (First paragraph, page 7) and figure legend. The description of the fraction of different types of vagal neurons in the main text does not match well with Figure 2, which includes several key elements. The authors should consider linking the data in the text more clearly to Figure 2.

The numbers in the main text included all compression and inflation-activated neurons, including multi-modal neurons, while the figure separated multi-modal neurons as a discrete response class. Also, the cell counts in the legend of Figure 2 mistakenly involved different stringency criteria- thank you for noticing this. We have now updated values to be consistent across main text, figure, and figure legend.

3. The authors might also want to describe how the cells (each row) in Figure 2b and Extended Data Figure 2a are grouped (manually or clustering using R) in the figure legend.

In the methods, we explain that for each responsive cell, the ratio (R_c/R_i) of response ($\Delta F/F_0$) to compression and inflation was calculated; cells were classified as compression-selective if $R_c/R_i > 2$, as inflation-selective if $R_c/R_i < 0.5$, or as polymodal if $0.5 < R_c/R_i < 2$. We now added an additional clarification that in Extended Data Figure 2a (now 2c, 2d), cells that did not respond to either airway inflation or airway closure were subsequently separated into two groups based on responsiveness to methacholine.

4. Based on the first paragraph of Page 8, Pvalb+ vagal neurons are a subtype of P2ry1+ vagal neurons. In authors' previous study, P2ry1+ vagal neurons including J3, NP16, NP17, NP19 and

NP26 clusters (Figure 2B of Prescott et al., 2020), two of which express Pvalb (NP17 & NP19) based on Figure 3c. It would be great if the authors could point out the specific Pvalb+ vagal type they are referring to. I speculate that NP17 and NP19 Pvalb+ neurons might have different functions.

Yes- we agree that there are other PVALB neurons in vagal ganglia, consistent with imaging data indicating that some but not all PVALB neurons respond to airway closure. We add a statement in the text saying, "airway closure activates 24.8% of vagal PVALB neurons, consistent with transcriptomic analysis indicating that the PVALB neurons constituting the transcriptome-defined cluster of gasp-promoting vagal sensory neurons represent 23.1% (161/698) of all PVALB neurons (Extended Data Fig. 3e)." We performed additional experiments in new Cre lines to further distinguish cell types relevant for gasping and swallowing. Our prior studies pinpointed NP19 neurons as relevant for swallowing. Finer analysis here reveals two NP19 neuron subtypes which we term here NP19a and NP19b. Optogenetics studies with new Cre lines show that NP19a neurons mediate swallowing while NP19b neurons (marked by Olf78) mediates gasping. We provide new analysis and data involving Olf78-Cre mice in Extended Data 3e.

5. In the discussion, the authors should recognize that these findings in mice have uncertain applicability to human and other large mammals both in terms of the distribution of NEBs and frequency of solitary neuroendocrine cells. It would also be very interesting to know whether PVALB+ (or CRHR2+) vagal neurons innervate human distal (bronchioles or alveolar) NEBs.

We agree that this is interesting to consider and add discussion of this point in the text. We note that we are not aware of a way to specifically label PVALB neurons in humans without genetic reagents.

6. Figure 3i, miss the label for the bottom right image.

Thank you for catching this!

7. In the last 2 lines of Page 11, the authors wrote that "Solitary neuroendocrine cells were only sparsely labeled in Piezo2-ires-Cre; Nkx2.1-Flpo mice". I cannot find such data. The extended data Figure 5a has an image of larynx from the Nkx2.1-Piezo2Cre mice, but I cannot see any fluorescent dots.

We have added quantification of labeled neuroendocrine cells by different allele combinations, and observed that Piezo2-ires-Cre; Nkx2.1-Flpo mice label less than 1% of these cells (Extended Data Fig 5b). We changed the text from 'were only sparsely labeled' to 'generally not labeled (<1%)' for clarity. This strengthens our claims of selectivity using these intersectional genetic tools.

8. The finding of PIEZO2 in NEBs (related to the first paragraph of Page 14 and Figure 5c). Recent gene expression profiling (Ouadah et al., 2019, Cell; Kuo et al. eLife 2022; LungMAP) of human

and mouse airways showed that neuroendocrine cells express PIEZO2 and other sensory receptors. In addition, in vitro physiological studies [Pan et al., 2006 (PMID: 16100287); Robrecht Lembrechts et al., 2012 (PMID: 22461428)] suggest that PNECs sense mechanical force. I would not directly claim that “underlying sensory receptors are unclear”. The knowledge gap should be the lack of physiological evidence in vivo.

Prior studies have made important observations about potential sensory receptors and sensory properties of NEBs, and we have cited each of these studies in the manuscript. Despite these advances, the roles of NEBs in physiology have remained mysterious, with roles in airway stretch sensation, hypoxia chemosensation, and immune surveillance previously put forth. Here, we have demonstrated a surprising role for NEBs in airway closure sensation, and have done so using a rigorous combination of complex genetic perturbations and physiological measurements. We refined the statement 'underlying sensory receptors are unclear' to "underlying sensory receptors that mediate airway closure are unclear".

9. The authors might move the heatmap in Figure 5b to the extended data. It is very interesting to show subtypes of neuroendocrine cells and their markers in a heatmap. The current heatmap does not include any gene names.

OK- we moved this panel to Extended Data Fig. 9b as suggested.

10. Figure 5d. I don't see quantification graphs (as figure legend claimed) for Piezo2 in situ hybridization.

Thank you for catching this. We now provide quantification of Piezo2-positive cells in Extended Data Fig. 10a.

11. In the extended data figure 8, I would not consider Epcam and Nkx2.1 as markers for the neuroendocrine cells since they are expressed in other epithelial cell types of the lungs based on other studies and single-cell RNA sequencing data.

We changed the figure legend so these are no longer described as NEB markers.

Reviewer Reports on the First Revision:

Referees' comments:

Referee #1 (Remarks to the Author):

The authors have done a thorough job addressing the key comments raised by reviewers and included impressive new imaging and recording/behavioral data that helps clarify the definition of 'gasp' and 'sigh'. I commend them on a remarkable paper, have no further questions/concerns, and look forward to seeing this published.

Referee #2 (Remarks to the Author):

No further comments. Great study. -Kevin

Referee #3 (Remarks to the Author):

The authors have addressed most of my comments. The manuscript is now in an excellent state for publication, and I only have a few minor suggestions for further refinement.

1. In text figure labels correlated to Extended Data Figure 9. 9c is the PNEC marker genes, 9d is the GO terms.
2. As I previously pointed out, EPCAM and NKX2.1 are not markers for PNECs. I am glad to see that the authors have corrected it in the figure legends. To maintain consistency and avoid confusion, I suggest removing the corresponding data in Extended Data Figure 9a (EPCAM column) and Extended Data Figure 9c (NKX2.1 and EPCAM plots).
3. I encourage the authors to enhance the heatmap (Extended Data Figure 9b) by labeling key marker genes associated with the identified PNEC subtypes. Or the authors could provide an extended table for the heatmap.
4. The authors should upload the single-cell transcriptomics data to the GEO database and share this information in the Data Availability section.

Author Rebuttals to First Revision:

Referee comments

Referee #1 (Remarks to the Author):

The authors have done a thorough job addressing the key comments raised by reviewers and included impressive new imaging and recording/behavioral data that helps clarify the definition of 'gasp' and 'sigh'. I commend them on a remarkable paper, have no further questions/concerns, and look forward to seeing this published.

Referee #2 (Remarks to the Author):

No further comments. Great study. -Kevin

Referee #3 (Remarks to the Author):

The authors have addressed most of my comments. The manuscript is now in an excellent state for publication, and I only have a few minor suggestions for further refinement.

1. In text figure labels correlated to Extended Data Figure 9. 9c is the PNEC marker genes, 9d is the GO terms.
2. As I previously pointed out, EPCAM and NKX2.1 are not markers for PNECs. I am glad to see that the authors have corrected it in the figure legends. To maintain consistency and avoid confusion, I suggest removing the corresponding data in Extended Data Figure 9a (EPCAM column) and Extended Data Figure 9c (NKX2.1 and EPCAM plots).
3. I encourage the authors to enhance the heatmap (Extended Data Figure 9b) by labeling key marker genes associated with the identified PNEC subtypes. Or the authors could provide an extended table for the heatmap.
4. The authors should upload the single-cell transcriptomics data to the GEO database and share this information in the Data Availability section.

We appreciate the helpful comments of all referees. We address the remaining minor suggestions of reviewer #3 here:

1. We edited the in-text callouts for the Extended Data Figure panels as suggested.
2. We removed the redundant EPCAM column in ED 9a, and further edited the ED9 figure legend.

3. We feel that it would be misleading to emphasize minor expression differences between NEB cells. Our principal conclusion is that NEBs serve a homogenous mechanosensory function based on single cell transcriptome data and functional data involving Piezo2 knockout. Subtle expression differences across NEB cells may not reflect functionally distinct subtypes with discrete sensory functions, but instead differences in cell state or development. We refer to a manuscript (Kuo et al, 2022) that nicely explores NEB heterogeneity in detail.

4. Yes- we will upload the transcriptomics dataset to GEO so it is freely available upon publication.